# Progressive-Hint Prompting Improves Reasoning in Large Language Models

Chuanyang Zheng [1]   Zhengying Liu [2]   Enze Xie [2]   Zhenguo Li [2]   Yu Li [1]

## Abstract

The performance of Large Language Models (LLMs) in reasoning tasks depends heavily on prompt design, with Chain-of-Thought (CoT) and self-consistency being critical methods that enhance this ability. However, these methods do not fully exploit the answers generated by the LLM to guide subsequent responses. This paper proposes a new prompting method, named Progressive-Hint Prompting (PHP), that enables automatic multiple interactions between users and LLMs by using previously generated answers as hints to progressively guide toward the correct answers. PHP is orthogonal to CoT and self-consistency, making it easy to combine with state-of-the-art techniques to further improve performance. We conducted extensive and comprehensive experiments on seven benchmarks. The results show that PHP significantly improves accuracy while remaining highly efficient. For instance, with text-davinci-003, we observed a 4.2% improvement on GSM8K with greedy decoding compared to Complex CoT, and a 46.17% reduction in sample paths with self-consistency. With GPT-4 and PHP, we achieve state-of-the-art performances on SVAMP (89.1% → 91.9%), GSM8K (92% → 95.5%), AQuA (76.4% → 79.9%) and MATH (50.3% → 53.9%).

## 1. Introduction

While Large Language Models (LLMs) have demonstrated remarkable performance across various NLP tasks (Otter et al., 2020; Qiu et al., 2020; Chowdhary & Chowdhary, 2020), their ability to reason is often perceived as a limitation that cannot be overcome merely by increasing the scale of the model (Rae et al., 2021; Srivastava et al., 2022).

Prompt engineering in large-scale models has shown comparable or superior performance to full training set fine-tuning in enhancing reasoning ability, while also being significantly more sample-efficient (Kojima et al., 2022; Lewkowycz et al., 2022). One area of research that aims to address this limitation is the use of Chain-of-Thought (CoT) approaches to promote intermediate reasoning steps (Wei et al., 2022; Zhou et al., 2023; Fu et al., 2023). Other works in this area, such as Least-to-Most (Zhou et al., 2023) and Complex CoT (Fu et al., 2023), have also explored this direction. Another area of research is self-consistency-related approaches. In comparison to CoT-related work that focuses on designing better prompts, self-consistency proposes to sample multiple answers from the LLMs and arrive at the correct answer through a majority vote (Fu et al., 2023). This approach is further improved upon by complex-based selection (Fu et al., 2023). CoT-related and self-consistency-related works can be seamlessly combined without any conflict.

Prior research has not explored the potential of leveraging the outputs of LLM to refine reasoning paths iteratively. It stands to reason that similar to human cognition, LLM could benefit from reevaluating and adjusting its generated reasoning paths in order to correct errors and enhance overall performance. In this paper, we propose a new method named Progressive-Hint Prompting (PHP) that involves sequentially interacting with LLM to approach the correct answer gradually. The method operates as follows: (1) given a question, we ask the LLM to provide a Base Answer; (2) we combine the question and answer to re-ask the LLM and obtain the Subsequent Answer; (3) we repeat the operation in (2) until the answer is stable and does not change over the last two answers. PHP follows a human-like thought process where previous answers are leveraged as hints to arrive at the correct answer after re-evaluating the question.

Figure 1 illustrates the proposed PHP framework. We use the base prompt to obtain the initial base answer, and then employ the PHP prompt for subsequent questions. If the current answer matches the previous answer, it is more likely to be correct, and we terminate the LLM inquiry. With Complex CoT and GPT-4, after adding PHP, the performance achieves SOTA with 91.9% on SVAMP (Patel et al., 2021), 95.5% on GSM8K (Cobbe et al., 2021), and 79.9% on

[1]The Chinese University of Hong Kong [2]Huawei Noah's Ark Lab. Correspondence to: Yu Li <liyu@cse.cuhk.edu.hk>.

*The first AI for MATH Workshop at the $41^{st}$ International Conference on Machine Learning*, Vienna, Austria. Copyright 2024 by the author(s).

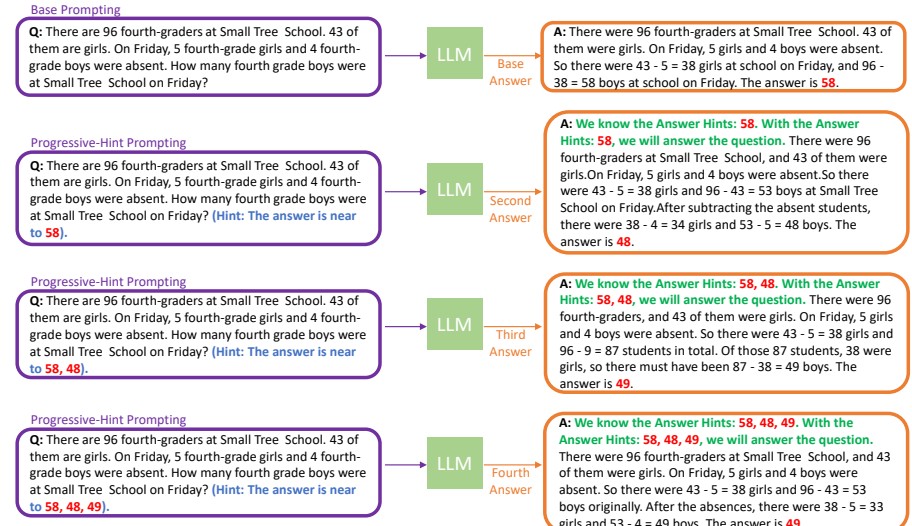

*Figure 1.* Our proposed Progressive-Hint Prompting method combines the generated answers and questions for double-checking purposes, which is divided into two stages. In the first stage, we generate a **base answer** by passing to the LLM a concatenation of the current question and a **base prompt**, such as CoT or Complex CoT. In the second stage, we generate the **subsequent answers** via the corresponding **progressive-hint prompt**, such as Progressive-Hint Prompting CoT (PHP-CoT) or Progressive-Hint Prompting Complex CoT (PHP-Complex CoT), for the subsequent interaction. The interaction stops when two consecutive answers are the same. Purple Box: The input of LLM. Orange Box: The output of LLM.

AQuA (Ling et al., 2017) and 53.9% on MATH (Hendrycks et al., 2021).

In summary, our contributions are as follows:

- We propose a new method, Progressive-Hint Prompting (PHP), alongside CoT and self-consistency, for improving LLM reasoning abilities.

- We demonstrate the effectiveness of PHP through extensive experimentation, including baseline comparisons and ablation studies, using four LLMs, text-davinci-002 and text-davinci-003, GPT-3.5-Turbo and GPT-4 (Brown et al., 2020; Ouyang et al., 2022; OpenAI, 2023).

- The experiment results show that our method can also improve performance with self-consistency.

- We believe that progressive-hint prompting represents an important step towards automatic sequential interaction with LLMs and hope that it inspires future research in this field.

## 2. Related Work

**Emergent Abilities and Multi-Step Reasoning.** LLMs are particularly skilled at in-context learning, which involves adhering to the structure of prompts (typically few-shot) and completing corresponding tasks (Brown et al., 2020; Chowdhery et al., 2022; Shin et al., 2020; Liu et al., 2023). Among the diverse range of language comprehension tasks, we are particularly interested in multi-step reasoning be-cause it exhibits two unique features. Firstly, LLMs significantly outperform smaller models on multi-step reasoning tasks (Wei et al., 2022), whereas their performance gains on tasks like sentiment classification can be limited (Shin et al., 2020). Secondly, few-shot prompting outperforms full training set fine-tuning in multi-step reasoning tasks, even when conducted on LLMs (Lewkowycz et al., 2022).

**Chain-of-Thought Reasoning.** Chain-of-thought (CoT) prompting (Wei et al., 2022) is a prominent work that demonstrates the multi-step reasoning capacities of LLMs. This approach suggests that the reasoning ability can be elicited through a chain of thoughts, where an answer directly follows a question without intermediate reasoning steps. Least-to-Most prompting (Zhou et al., 2023), which follows the same research direction, divides reasoning into problem breakdown parts and problem answer parts and describes the reasoning steps in more detail. Similarly, the complex CoT (Fu et al., 2023) highlights the importance of prompt complexity and selects the most complex questions and their answers as prompts. To reduce the human workload, the Auto-CoT is proposed (Zhang et al., 2022). Other works have found that using specific phrases like "Let's think step by step" (Kojima et al., 2022) can improve performance.

**Reasoning Path Extraction.** Previous research has investigated various task-specific methods for identifying reasoning paths, including constructing semantic graphs (Xu et al., 2021), developing Recurrent Neural Network (RNN) models to retrieve reasoning paths from a Wikipedia graph (Asai

et al., 2020), using human-annotated reasoning paths on math problems for fine-tuning (Cobbe et al., 2021), or training an extractor with heuristic-based pseudo reasoning paths (Chen et al., 2019). A novel research work, named Self-Consistency (Wang et al., 2023), couples the generation of reasoning paths and a final answer by sampling from the decoder and using aggregation to retrieve the most consistent answer without extra modules. This approach has shown great promise, and it has the potential to outperform existing methods in terms of accuracy. Furthermore, the vote complex (Fu et al., 2023), rank and select samples via complexity, is proposed to improve the self-consistency performance. This approach is particularly useful when dealing with complex reasoning problems.

## 3. Progressive-Hint Prompting

_Table 1._ Illustration of Progressive-Hint Prompting. Blue Color: The difference between Base CoT and PHP-CoT. Red Color: The handcrafted Hint in the designed prompt.

**Base Prompting (_e.g._ CoT)**

Q: There are 15 trees in the grove. Grove workers will plant trees in the grove today. After they are done, there will be 21 trees. How many trees did the grove workers plant today?
A: There are 15 trees originally. Then there were 21 trees after the Grove workers planted some more. So there must have been 21 - 15 = 6 trees that were planted. The answer is 6.

**Progressive-Hint Prompting 1: Hint is the correct answer**

Q: There are 15 trees in the grove. Grove workers will plant trees in the grove today. After they are done, there will be 21 trees. How many trees did the grove workers plant today? (Hint: The answer is near to 6).
A: We know the Answer Hints: 6. With the Answer Hints: 6, we will answer the question. There are 15 trees originally. Then there were 21 trees after the Grove workers planted some more. So there must have been 21 - 15 = 6 trees that were planted. The answer is 6.

**Progressive-Hint Prompting 2: Hint is the incorrect answer**

Q: There are 15 trees in the grove. Grove workers will plant trees in the grove today. After they are done, there will be 21 trees. How many trees did the grove workers plant today? (Hint: The answer is near to 10, 8).
A: We know the Answer Hints: 10, 8. With the Answer Hints: 10, 8, we will answer the question. There are 15 trees originally. Then there were 21 trees after the Grove workers planted some more. So there must have been 21 - 15 = 6 trees that were planted. The answer is 6.

One salient aspect of humanity is our ability to not only think once, but to also double-check our answers. In this paper, we propose that this process can be simulated in language models by sequentially employing previous answers. In other words, a model can generate an answer and then combine this with the question for the next round of thinking. If the current answer is the same as the previous one, we can have confidence that the current answer is correct.

We have shown the Proposed Interaction in Figure 1 and Prompt Design in Table 1. We demonstrate the process of generating PHP-CoT prompts for a given CoT prompt in Table 1 and provide the complete prompt in the Appendix. Our pipeline is divided into two stages: (i) **base answer & base prompt**: the generation of the base answer via base prompts such as CoT or Complex CoT and (ii) **subsequent answer & PHP**: the subsequent interaction with the LLMs through corresponding progressive-hint prompts like Progressive-Hint Prompting CoT (PHP-CoT) or Progressive-Hint Prompting Complex CoT (PHP-Complex CoT). We propose a two-sentence structure for the PHP, consisting of a phrase indicating the proximity of the answer at the question part followed by a sentence rehearsing hints at answer part. For instance, to create a PHP prompt from a CoT prompt, we first add the phrase "The answer is near to $A_1, ..., A_p$" after the initial question, where $A_1, ..., A_p$ represent possible answers. Next, we introduce the hints in the beginning sentence of the potential answers: "We know the Answer Hints: $A_1, ..., A_p$. With the Answer Hints: $A_1, ..., A_p$, we will answer the question.".

**PHP Design Principle**: we should consider various situations of hints. When we ask LLM questions, we do not know what the answer will be so the hints are unknown. In this prompt design, we consider the following two potential situations: 1) The hints are the same as the correct answer: to be sure that the model can still get the correct answer when the hint is correct; 2) hints are not the same as the correct answer: to be sure that the model can jump out of the incorrect answer.

Adhering to the above guidelines, we utilize the Standard prompt, CoT prompt, and Complex CoT prompt to generate initial base answers, from which we can then develop the subsequent answer generation prompts, namely, **PHP-Standard** prompt, **PHP-CoT** prompt, and **PHP-Complex CoT** prompt, respectively. The stopping criterion in PHP is reached when two consecutive responses are identical, signaling the end of the interactive exchange.

Overall, this method represents a pipeline for improving the quality of responses and enhancing communication during question-answer scenarios.

# 4. Experiments

**Datasets and Models.** We evaluate PHP on seven datasets (AddSub (Hosseini et al., 2014), MultiArith (Roy & Roth, 2015), SingleEQ (Koncel-Kedziorski et al., 2015), SVAMP (Patel et al., 2021), GSM8K (Cobbe et al., 2021), AQuA (Ling et al., 2017)) and MATH (Hendrycks et al., 2021). We choose the datasets because we focus on the reasoning ability of the model. The utilized for both the Standard and CoT prompts are sourced from the original CoT paper (Wei et al., 2022), whereas the prompt utilized for the Complex CoT (Fu et al., 2023) prompt is derived from the corresponding Complex CoT publication. Also, to validate our proposed method performance, we employ four models: text-davinci-002 and text-davinci-003, GPT-3.5-Turbo and GPT-4 (Brown et al., 2020; Ouyang et al., 2022; OpenAI, 2023). All models are employed via OpenAI API key.

**Prompts.** We have shown the proposed process pipeline in the Method part. We show all the prompts in the Appendix and supplementary materials.

## 4.1. Main Results

The main results of our study are presented in Table 2, with all methods using greedy decoding (i.e. temperature = 0). Our findings indicate that the proposed PHP improves performance, particularly when working with powerful prompts and models.
**PHP works better when the LLM is more powerful**. In terms of model power, our analysis indicates that PHP is most effective when applied with powerful models. Specifically, when examining CoT and Complex CoT prompts, we found that while text-davinci-002 generally yielded a performance improvement after adding hints, there were occasions when performance would decline. However, when we replaced text-davinci-002 with text-davinci-003, performance improvement became more consistent and significant. For example, on GSM8K dataset, PHP-Complex CoT using text-davinci-002 improved performance by 3.6%, but then increased further to 4.6% with text-davinci-003. Similarly, on AQuA dataset, using PHP-Complex CoT resulted in a performance drop of 0.4% with text-davinci-002 but a 1.2% improvement with text-davinci-003. The text-davinci-002 is finetuned with supervised instruction tuning, while the text-davinci-003 is finetuned with reinforcement learning. The improved performance with text-davinci-003 can be attributed to its enhanced power, making it better at understanding and employing the given hint.

**PHP works better when the prompt is more powerful.** After analyzing our data, it was determined that the prompt's power has a significant impact on the performance of the system. Our experimental results revealed that while the inclusion of PHP produced modest improvements with standard

*Table 2.* PHP, when applied to different LLMs and prompting methods, can help to improve the performance. Meanwhile, PHP works better when the model and prompt are more powerful. The results are with greedy decoding.

| Prompt | PHP | AddSub | MultiArith | SingleEQ | SVAMP | GSM8K | AQuA | Average |
|---|---|---|---|---|---|---|---|---|
| **GPT-3.5 text-davinci-002** | | | | | | | | |
| Standard (Wei et al., 2022) | ✗ | 79.4 | 34.0 | 80.7 | 64.8 | 15.1 | 25.5 | 49.91 |
| | ✓ | 80.5 | 31.8 | 79.9 | 64.2 | 14.7 | 25.5 | 49.43 |
| | | (+1.1) | (-2.2) | (-0.8) | (-0.6) | (-0.4) | (0.0) | (-0.48) |
| CoT (Wei et al., 2022) | ✗ | 85.8 | 89.1 | 89.7 | 72.9 | 49.5 | 44.4 | 71.89 |
| | ✓ | 86.8 | 89.0 | 90.1 | 72.3 | 51.1 | 45.6 | 72.48 |
| | | (+1.0) | (-0.1) | (+0.4) | (-0.6) | (+1.6) | (+1.2) | (+0.59) |
| Complex CoT (Fu et al., 2023) | ✗ | 82.5 | 89.8 | 87.7 | 70.4 | 57.6 | 37.4 | 70.89 |
| | ✓ | 83.7 | 90.1 | 89.9 | 74.6 | 61.2 | 37.0 | 72.75 |
| | | (+1.2) | (+0.3) | (+2.2) | (+4.2) | (+3.6) | (-0.4) | (+1.86) |
| **GPT-3.5 text-davinci-003** | | | | | | | | |
| Standard (Wei et al., 2022) | ✗ | 89.1 | 36.3 | 83.8 | 68.7 | 15.9 | 28.3 | 53.68 |
| | ✓ | 89.1 | 36.0 | 83.6 | 68.7 | 16.0 | 28.3 | 53.61 |
| | | (0.0) | (-0.3) | (-0.2) | (0.0) | (+0.1) | (0.0) | (-0.07) |
| CoT (Wei et al., 2022) | ✗ | 90.6 | 93.6 | 92.7 | 81.0 | 56.1 | 44.0 | 76.33 |
| | ✓ | 91.1 | 94.0 | 93.5 | 81.3 | 57.5 | 44.4 | 76.96 |
| | | (+0.5) | (+0.4) | (+0.8) | (+0.3) | (+1.4) | (+0.4) | (+0.63) |
| Complex CoT (Fu et al., 2023) | ✗ | 86.3 | 94.8 | 91.5 | 77.4 | 67.0 | 48.8 | 77.63 |
| | ✓ | 88.1 | 95.0 | 94.0 | 80.0 | 71.6 | 50.0 | 79.78 |
| | | (+1.8) | (+0.2) | (+2.5) | (+2.6) | (+4.6) | (+1.2) | (+2.15) |

prompts, CoT and Complex CoT prompts demonstrated substantial gains in performance. Particularly noteworthy is the fact that the most potent prompt, Complex CoT, exhibited the most substantial performance improvement in comparison to the Standard prompt and CoT prompt. The in-context learning imparts a pattern to the model, and the quality of the prompt directly influences the model's ability to learn from this pattern. As indicated by the experiments in Table 2, the Complex CoT prompt outperforms the CoT prompt, and the CoT prompt surpasses the Standard prompt. Consequently, it is more advantageous for the Complex CoT to instruct the LLM in pattern recognition. Within the proposed PHP framework, the established pattern is as follows: 1) if the initial answer is correct, maintain the same correct response in the subsequent round; 2) if the initial answer is incorrect, strive to provide the correct answer in the next round. The Standard prompt falls short of effectively instilling such a pattern in the model, resulting in minimal variation in LLM's responses and a reduced number of interactions. In contrast, the Complex CoT excels in instructing the LLM to rectify its responses, facilitating a more dynamic and responsive learning process. This finding provides compelling evidence that a superior prompt leads to greater effectiveness of the system.

**The Interaction Number decreases when the model is more powerful and the prompt is less powerful.** The number of interactions refers to how many times the agent engages with the LLMs. The interaction number is one when the agent receives the first answer and increases to two for the second answer. In Figure 2, we illustrate the interaction number of various models and prompts. Our findings indicate that: 1) The interaction number for text-davinci-003 is typically lower than that of text-davinci-002 when given the same prompt. This is primarily due to the higher accuracy of text-davinci-003, resulting in a higher probability of the base answer and subsequent answers being correct, thus requiring fewer interactions to obtain the final correct answer; 2) When using the same models, the in-

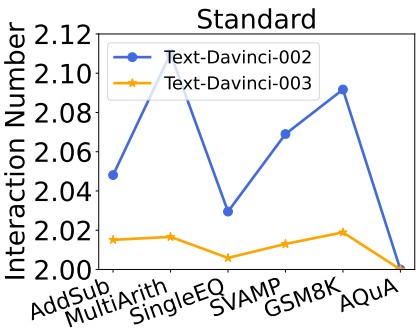
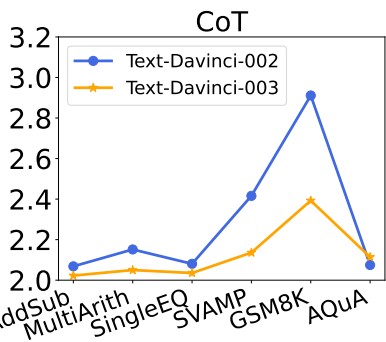
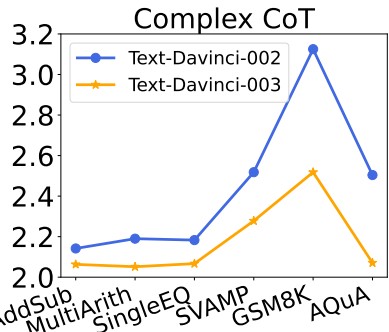

*Figure 2.* The Interaction Number refers to the frequency at which we need to consult the LLM until we receive conclusive responses. With an analysis of various models and prompts, it has been observed that: 1) A stronger model leads to a decreased interaction number; 2) An improved prompt results in an increased interaction number.

teraction number generally increases as the prompt becomes more powerful. This is because the LLMs achieves better reasoning ability when the prompt becomes more potent, allowing them to leverage the hints to jump out of the incorrect answers, and ultimately leading to a higher number of interactions required to reach the final answer.

## 4.2. Impact of the Hint Quality

*Table 3.* Performance with different Base Answers. Initially, the base prompt provides base answers to the model and PHP generates the subsequent answers. The results are from text-davinci-003 with greedy decoding.

| PHP Prompt | Base Prompt | Dataset | | | | | | Average |
|---|---|---|---|---|---|---|---|---|
| | | AddSub | MultiArith | SingleEQ | SVAMP | GSM8K | AQuA | |
| PHP-Standard | Standard (Wei et al., 2022) | 89.1 | 36.0 | 83.6 | 68.7 | 16.0 | 28.3 | 53.61 |
| | CoT (Wei et al., 2022) | **92.4** | 80.5 | 92.1 | **78.5** | 50.2 | 42.5 | 72.70 |
| | Complex CoT (Fu et al., 2023) | 90.6 | **80.6** | **92.9** | 77.2 | **60.3** | **45.6** | **74.53** |
| PHP-CoT | Standard (Wei et al., 2022) | 90.8 | 92.5 | 90.7 | 80.2 | 52.3 | 40.9 | 74.56 |
| | CoT (Wei et al., 2022) | **91.1** | 94.0 | 93.5 | **81.3** | 57.5 | 44.4 | 76.96 |
| | Complex CoT (Fu et al., 2023) | 90.6 | **96.8** | **93.7** | 81.2 | **62.6** | **50.0** | **79.14** |
| PHP-Complex CoT | Standard (Wei et al., 2022) | 88.3 | 80.1 | 93.3 | 80.4 | 65.5 | 35.4 | 73.83 |
| | CoT (Wei et al., 2022) | **88.8** | **95.6** | **94.8** | **81.4** | 70.6 | 45.6 | 79.46 |
| | Complex CoT (Fu et al., 2023) | 88.1 | 95.0 | 94.0 | 80.0 | **71.6** | **50.0** | **79.78** |

**The quality of the hint significantly affects the performance.** Shown in Table 3, to enhance the PHP-Standard, replacing the base prompt Standard with Complex CoT or CoT leads to a significant improvement in the final performance. For PHP-Standard, we observe that GSM8K performance amplifies from 16.0% with base prompt Standard to 50.2% with base prompt CoT and 60.3% with base prompt Complex CoT. Conversely, replacing the base prompt Complex CoT with Standard will reduce the final performance. For example, after replacing base prompt Complex CoT with Standard, the performance of PHP-Complex CoT drops from 71.6% to 65.5% on GSM8K dataset.

**Performance may further improve if PHP is not designed from the corresponding base prompt.** The results indicate that the CoT with PHP-Complex CoT achieved a high accuracy rate of 96.8% on the MultiArith dataset, surpassing the performance of the CoT with PHP-CoT. Similarly,

the Complex CoT with PHP-CoT demonstrated a notable accuracy rate of 95.6% on the same dataset, outperforming the Complex CoT with PHP-Complex CoT. The rationale behind these findings is twofold: 1) the performance of CoT and Complex CoT are similar on all six datasets, and 2) since the Base answer is provided by CoT (or Complex CoT) and the subsequent answer is based on PHP-Complex CoT (or PHP-CoT), it is comparable to having two individuals collaborating to solve a problem. Therefore, in such circumstances, the system's performance may be further enhanced.

## 4.3. Ablation Study

Furthermore, we conducted an ablation study to verify the criticality of the two sentences in answers: 1) P1: We know the Answer Hints $A_1, ..., A_p$; 2) P2: With the Answer Hints $A_1, ..., A_p$, we will answer the question. Moreover, we introduced a new type of prompt called CoT-Merge and Complex CoT-Merge. Firstly, we combined the original prompt with the PHP prompt into a single file. Subsequently, we utilized the same Merge Prompt for both the base answer and subsequent answers. Also, we prove that both correct and incorrect hints are necessary for prompt design. We employ the stop criterion (adaptive sampling) to determine termination for all experiments.

**The Proposed P1 and P2 are necessary.** Incorporating the sentences P1 and P2 resulted in better performance for CoT with PHP across three of the six datasets. However, the significance of these two sentences became particularly apparent when we employed Complex CoT. With this method, better performance was achieved on five of the six datasets after adding P1 and P2. For instance, Complex CoT improved its performance from 78.0% to 80.0% on the SVAMP dataset, and from 68.3% to 71.6% on the GSM8K dataset. This highlights that sentences P1 and P2 can exhibit more potent abilities, particularly when the model's logical capac-

*Table 4.* Ablation Study. CoT-Merge: for the CoT base prompt and the PHP-CoT prompt, we employ the prompt that contains both base prompt and the PHP. **P1**: We know the Answer Hints $A_1, ..., A_p$. **P2**: With the Answer Hints $A_1, ..., A_p$, we will answer the question. According to the experiment results, we see that both the proposed P1 and P2 are necessary. Meanwhile, non-merge based method is better than merge based method when prompts are more powerful. The results are from text-davinci-003 with greedy decoding.

| Base Prompt | PHP Prompt | P1 | P2 | Dataset | | | | | | Average |
|---|---|---|---|---|---|---|---|---|---|---|
| | | | | AddSub | MultiArith | SingleEQ | SVAMP | GSM8K | AQuA | |
| CoT | N/A | N/A | N/A | 90.6 | 93.6 | 92.7 | 81.0 | 56.1 | 44.0 | 76.33 |
| CoT-Merge | CoT-Merge | ✓ | ✓ | **91.3** | **94.6** | 93.1 | 79.5 | 58.6 | **50.0** | **77.85** |
| CoT | PHP-CoT | ✗ | ✗ | 91.1 | 93.5 | 93.3 | 80.0 | 58.1 | 44.8 | 76.80 |
| | | ✓ | ✗ | 90.8 | 93.1 | 92.9 | 80.7 | **58.8** | 43.7 | 76.66 |
| | | ✗ | ✓ | **91.3** | 93.8 | 93.5 | 80.5 | 58.2 | 46.4 | 77.28 |
| | | ✓ | ✓ | 91.1 | 94.0 | **93.5** | **81.3** | 57.5 | 44.4 | 76.96 |
| Complex CoT | N/A | N/A | N/A | 86.3 | 94.8 | 91.5 | 77.4 | 67.0 | 48.8 | 77.63 |
| Complex CoT-Merge | Complex CoT-Merge | ✓ | ✓ | **88.8** | 94.3 | 94.6 | 78.1 | 70.2 | 46.8 | 78.80 |
| Complex CoT | Complex CoT | ✗ | ✗ | 87.8 | 93.3 | 93.7 | 78.0 | 68.3 | **50.3** | 78.56 |
| | | ✓ | ✗ | 87.8 | **95.1** | 94.2 | 78.5 | 70.5 | 48.4 | 79.08 |
| | | ✗ | ✓ | 88.3 | 94.3 | **94.6** | 79.1 | 69.3 | 46.8 | 78.73 |
| | | ✓ | ✓ | 88.1 | 95.0 | 94.0 | **80.0** | **71.6** | 50.0 | **79.78** |

*Table 5.* Analysis of Hint Design (Shown in Figure 1). Correct: The hints of designed prompt are the same as the correct answers. Incorrect: The hints of the designed prompt are the incorrect answers. Green: The performance is better than without progressive-hint. Red: The performance is worse than without progressive-hint. The results are from text-davinci-003 with greedy decoding.

| Method | Hint | | Dataset | | | | | | Average |
|---|---|---|---|---|---|---|---|---|---|
| | Correct | Incorrect | AddSub | MultiArith | SingleEQ | SVAMP | GSM8K | AQuA | |
| CoT (Wei et al., 2022) | ✗ | ✗ | 90.6 | 93.6 | 92.7 | 81.0 | 56.1 | 44.0 | 76.33 |
| | ✓ | ✗ | 91.6 | 94.3 | 93.3 | 81.9 | 57.0 | 43.7 | 76.96 |
| | ✗ | ✓ | 91.1 | 93.5 | 93.1 | 79.7 | 57.9 | 45.2 | 76.74 |
| | ✓ | ✓ | 91.1 | 94.0 | 93.5 | 81.3 | 57.5 | 44.4 | **76.96** |
| Complex CoT (Fu et al., 2023) | ✗ | ✗ | 86.3 | 94.8 | 91.5 | 77.4 | 67.0 | 48.8 | 77.63 |
| | ✓ | ✗ | 88.3 | 94.0 | 93.8 | 77.8 | 68.6 | 46.4 | 78.14 |
| | ✗ | ✓ | 88.1 | 94.6 | 94.0 | 79.2 | 70.2 | 48.4 | 79.08 |
| | ✓ | ✓ | 88.1 | 95.0 | 94.0 | 80.0 | 71.6 | 50.0 | **79.78** |

ity is superior. Consequently, we can conclude that P1 and P2 will likely enhance model performance to a greater extent, particularly with more powerful prompts and models.

**Non-Merge based PHP is better than merge based PHP when prompts are more powerful.** Regarding the CoT with PHP-CoT, the initial answer is derived from the CoT prompt, and subsequently, the answer is obtained from the PHP-CoT. Notably, compared to other CoT-base methods, CoT-Merge achieves the best performance. However, compared to other Complex CoT-based methods, we observe that non-merge PHP-Complex CoT with both P1 and P2 achieves the best performance. Hence, when prompts are better, the performance of non-merge based method will be better than merge-based method.

**Both correct and incorrect hints are needed in the prompt design.** Table 5 demonstrates that the use of PHP was superior to its absence when the designed prompt included both correct and incorrect hints. Specifically, the provision of a correct hint in the prompt promoted the generation of answers that matched the given hint. Conversely, the provision of incorrect answers in the prompt encouraged the generation of alternative answers, with the aid of the given hint.

*Table 6.* The results after adding Self-Consistency (SC). **Number**: The interaction number between agent and LLM. The best results of adding PHP are highlighted with red color, and the best results without PHP are highlighted with green color. We find that PHP further improves performance, even adding self-consistency. Meanwhile, PHP may reduce the cost of self-consistency.

| Prompt | SC | PHP | Dataset | | | | | | Average |
|---|---|---|---|---|---|---|---|---|---|
| | | | AddSub | MultiArith | SingleEQ | SVAMP | GSM8K | AQuA | |
| CoT (Wei et al., 2022) | 5 | ✗ | 90.6 | 95.3 | 94.4 | 81.6 | 63.3 | 49.2 | 79.06 |
| | 5 | ✓ | 90.8 | 96.6 | 94.8 | 83.5 | 66.3 | 49.6 | 80.26 |
| | 5 | Number | 2.0075 | 2.0433 | 2.0098 | 2.1090 | 2.5458 | 2.0157 | 2.1218 |
| | 10 | ✗ | 90.6 | 96.5 | 93.8 | 83.0 | 65.5 | 49.2 | 79.76 |
| | 10 | ✓ | 90.8 | 97.1 | 93.8 | 83.5 | 67.5 | 50.0 | 80.45 |
| | 10 | Number | 2.0075 | 2.0283 | 2.0059 | 2.0510 | 2.2145 | 2.0118 | 2.0531 |
| | 20 | ✗ | 91.1 | 96.5 | 94.2 | 83.3 | 68.0 | 55.1 | 81.36 |
| | 20 | ✓ | 91.6 | 96.5 | 94.4 | 83.7 | 68.6 | 55.1 | 81.64 |
| | 20 | Number | 2.0050 | 2.0366 | 2.0098 | 2.0250 | 2.1144 | 2.0078 | 2.0330 |
| | 40 | ✗ | 91.6 | 96.5 | 94.8 | 82.9 | 67.3 | 53.1 | 81.03 |
| | 40 | ✓ | 91.6 | 96.6 | 95.0 | 83.7 | 68.4 | 53.1 | 81.39 |
| | 40 | Number | 2.0050 | 2.0300 | 2.0050 | 2.0320 | 2.0530 | 2.0000 | 2.0208 |
| Complex CoT (Fu et al., 2023) | 5 | ✗ | 88.1 | 97.0 | 93.1 | 80.4 | 73.5 | 51.5 | 80.60 |
| | 5 | ✓ | 89.6 | 97.3 | 95.2 | 82.5 | 76.9 | 51.9 | 82.23 |
| | 5 | Number | 2.0378 | 2.0166 | 2.0334 | 2.2370 | 2.5390 | 2.0118 | 2.1459 |
| | 10 | ✗ | 88.6 | 98.3 | 93.3 | 82.4 | 76.4 | 54.3 | 82.21 |
| | 10 | ✓ | 89.1 | 98.5 | 95.2 | 83.4 | 78.2 | 54.7 | 83.18 |
| | 10 | Number | 2.0177 | 2.0016 | 2.0295 | 2.059 | 2.1531 | 2.0078 | 2.0447 |
| | 20 | ✗ | 88.6 | 98.0 | 93.8 | 82.5 | 77.7 | 56.2 | 82.80 |
| | 20 | ✓ | 89.8 | 98.0 | 95.8 | 83.6 | 78.6 | 56.2 | 83.66 |
| | 20 | Number | 2.0253 | 2.0000 | 2.0196 | 2.0330 | 2.0401 | 2.0000 | 2.0196 |
| | 40 | ✗ | 88.3 | 98.5 | 94.8 | 83.9 | 78.1 | 58.6 | 83.70 |
| | 40 | ✓ | 88.6 | 98.5 | 95.8 | 84.7 | 79.0 | 58.6 | 84.20 |
| | 40 | Number | 2.0101 | 2.0000 | 2.0137 | 2.0210 | 2.0348 | 2.0039 | 2.0137 |

## 4.4. Performance with Self-Consistency

As we discussed before, our proposed method can combine with CoT and self-consistency to further improve the model performance. The results are shown in Table 6. Following the self-consistency paper, we sample paths with numbers 5, 10, 20 and 40, and the model temperature 0.7.

**PHP further improves performance.** By utilizing similar prompts and sample path numbers, we discovered that our proposed PHP-CoT and PHP-Complex CoT always achieve superior performance when compared to CoT and Complex CoT, shown in Table 6 and Figure 3. For instance, CoT with self-consistency was able to attain a 96.5% accuracy on the MultiArith dataset with a sample path of 10, 20 and 40. Therefore, we can conclude that the best performance by CoT with self-consistency is 96.5% with text-davinci-003. However, after implementing PHP, the performance skyrocketed to 97.1%. Similarly, we observed CoT with self-consistency on SVAMP, achieving the best accuracy of 83.3% with 20 sampled paths, and further improved to 83.7% upon implementing PHP. This illustrates that PHP could break the performance bottleneck and further improve performance.

**PHP could reduce the cost of self-consistency.** Incorporating the PHP can also lead to cost reduction. It is widely acknowledged that self-consistency involves an increased number of reasoning paths, resulting in a higher cost. The Table 6 illustrates that PHP can be an effective approach for reducing this cost, while still preserving performance gains. As shown in Figure 3, using Complex CoT with self-consistency, a 78.1% accuracy can be reached with 40 sample paths, while incorporating PHP reduces the required sample amount to $10 \times 2.1531 = 21.531$ paths, and results in

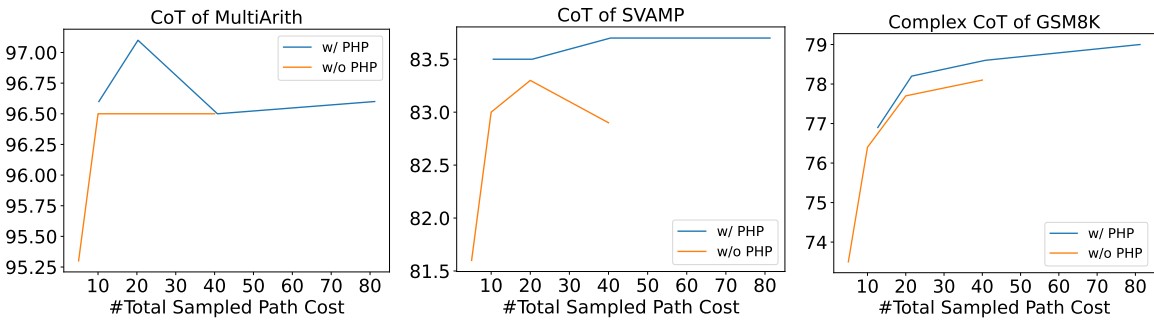

*Figure 3.* We show the results: 1) CoT of MultiArith; 2) CoT of SVAMP; 3) Complex CoT of GSM8K. According to 1) and 2), we can see that PHP could further improve performance. With result 3), we found that the PHP could even reduce the cost of self-consistency.

an even better accuracy of 78.2%.

## 4.5. Performance with Chat Model

*Table 7.* Performance of Complex CoT with GPT-3.5-Turbo and GPT-4, employing greedy decoding. Number: The average interaction number with LLM.

| | PHP | | | | Dataset | | | Average |
|---|---|---|---|---|---|---|---|---|
| | | AddSub | MultiArith | SingleEQ | SVAMP | GSM8K | AQuA | |
| Previous SOTA | | 94.9 (Roy & Roth, 2015) | 100 (Wang et al., 2023) | 95.5 (Diao et al., 2023) | 89.1 (Chen et al., 2022) | 92.0 (OpenAI, 2023) | 76.4 (Pitis et al., 2023) | 91.31 |
| GPT-3.5 Turbo | ✗ | 85.5 | 97.5 | 92.5 | 81.0 | 82.8 | 57.4 | 82.78 |
| | ✓ | 85.3 | 98.0 | 92.9 | 83.1 | 85.1 | 60.6 | 84.16 |
| | | (-0.2) | (+0.5) | (+0.4) | (+2.1) | (+2.3) | (+3.2) | (+1.38) |
| | Number | 2.1037 | 2.0133 | 2.0610 | 2.3570 | 2.3426 | 2.3228 | 2.2000 |
| GPT-4 | ✗ | 89.3 | 97.8 | 93.1 | 90.5 | 94.9 | 77.5 | 90.51 |
| | ✓ | 89.6 | 98.1 | 93.1 | 91.9 | 95.5 | 79.9 | 91.34 |
| | | (+0.3) | (+0.3) | (0.0) | (+1.4) | (+0.6) | (+2.4) | (+0.83) |
| | Number | 2.0126 | 2.0033 | 2.0019 | 2.0700 | 2.0507 | 2.2913 | 2.0716 |

In the previous section, we follow the previous work settings and employ text generation models for our experiments. With the release API of GPT-3.5-Turbo and GPT-4, we validate the performance of Complex CoT with PHP on the same six datasets. We use greedy decoding (i.e. temperature = 0) and Complex CoT as prompt for both models.

**Analyze GPT-3.5-Turbo.** Let's delve into an in-depth analysis of GPT-3.5-Turbo, as detailed in Table 7. Our proposed PHP exhibits remarkable performance enhancements, resulting in a substantial 2.3% improvement on GSM8K and an impressive 3.2% boost on AQuA. However, it's worth noting that GPT-3.5-Turbo appears to have a diminished capability when it comes to adhering to prompts compared to its counterpart, text-davinci-003. To illustrate this disparity, we can provide two concrete examples: a) In scenarios where the provided hints are conspicuously absent, GPT-3.5-Turbo encounters difficulties in providing an answer, often responding with a statement such as, "We cannot answer this question as the answer hint is missing. Please provide the answer hint to proceed." On the other hand, text-davinci-003 autonomously generates and fills in the missing answer hint before addressing the question, a phenomenon that is well-demonstrated in Table 8. b) When confronted with more than ten hints, GPT-3.5-Turbo may respond with the message, "We cannot determine the correct answer as multiple answer hints are given. Please provide only one answer hint for the question." Also, Such behavior is not observed

*Table 8.* Response when miss hint for GPT-3.5 Text-Divinci-003, with Complex CoT

Question: Jerry has two pools, both with leaks emptying them out at 4 gallons/minute. 4 minutes ago, the big pool had twice as much water as the small pool. Now the big pool has four times as much water as the small pool. How much water does the small pool have now? (Hint: The answer is near to ).

A: We know the Answer Hints: 16. With the Answer Hints: 16, we will answer the question.
Let's think step by step
4 minutes ago, the big pool had twice as much water as the small pool.
This means that the small pool had 4 gallons of water.
In 4 minutes, the big pool has lost 4 gallons x 4 minutes = 16 gallons of water.
The small pool has also lost 4 gallons x 4 minutes = 16 gallons of water.
Therefore, the small pool now has 4 gallons - 16 gallons = -12 gallons of water.
Since it is impossible to have negative gallons of water, the small pool must have 0 gallons of water.
The answer is 0.

in text-davinci-003. Consequently, OpenAI may employ different implement alternative techniques to grant GPT-3.5-Turbo more response flexibility. As a result, the model's responses occasionally deviate from the given prompt, compared to those generated by GPT-3.5 text-davinci-0003.

**Analyze GPT-4.** The GPT-4 model has significantly improved performance, showcasing its effectiveness across various benchmarks. It has achieved high accuracy rates: 91.9% on SVAMP, 95.5% on GSM8K, 79.9% on AQuA, and 53.90% on the challenging MATH dataset. These results are a testament to the effectiveness of our PHP methodology, which has been instrumental in enhancing GPT-4's capabilities. Particularly notable is the improvement on the MATH dataset, where PHP increased accuracy from

*Table 9.* Performance of Complex CoT with GPT-3.5-Turbo and GPT-4 on MATH dataset, employing greedy decoding. Number: The average interaction number with LLM. **Overall**: The results overall MATH subtopics (Hendrycks et al., 2021).

| | PHP | MATH Dataset | | | | | | | |
|---|---|---|---|---|---|---|---|---|---|
| | | InterAlgebra | Precalculus | Geometry | NumTheory | Probability | PreAlgebra | Algebra | Overall |
| Previous SOTA(Lewkowycz et al., 2022) | ✗ | - | - | - | - | - | - | - | 50.30 |
| GPT-4 CoT(OpenAI, 2023) | ✗ | - | - | - | - | - | - | - | 42.50 |
| GPT-3.5-Turbo Complex CoT (Ours) | ✗ | 14.6 | 16.8 | 22.3 | 33.4 | 29.7 | 53.8 | 49.1 | 34.12 |
| | ✓ | 17.1 | 16.1 | 25.4 | 35.1 | 33.7 | 57.7 | 51.1 | 36.50 |
| | | (+2.5) | (-0.7) | (+3.1) | (+1.7) | (+4.0) | (+3.9) | (+2.0) | (+2.38) |
| | Number | 4.2746 | 3.9625 | 4.3361 | 3.8166 | 3.7594 | 3.1526 | 3.0716 | 3.6673 |
| GPT-4 Complex CoT (Ours) | ✗ | 23.4 | 26.7 | 36.5 | 49.6 | 53.1 | 71.6 | 70.8 | 50.36 |
| | ✓ | 26.3 | 29.8 | 41.9 | 55.7 | 56.3 | 73.8 | 74.3 | 53.90 |
| | | (+2.9) | (+3.1) | (+5.4) | (+6.1) | (+3.2) | (+2.2) | (+3.5) | (+3.54) |
| | Number | 3.2414 | 3.2435 | 3.2233 | 3.1740 | 2.8122 | 2.3226 | 2.4726 | 2.8494 |

*Table 10.* The interaction number distribution of different datasets, with GPT-4 and Complex CoT.

| Interaction Number | Dataset | | | | | |
|---|---|---|---|---|---|---|
| | AddSub | MultiArith | SingleEQ | SVAMP | GSM8K | AQuA |
| 2 | 98.98% | 99.66% | 99.80% | 95.80% | 97.42% | 84.64% |
| 3 | 0.75% | 0.33% | 0.19% | 2.80% | 1.44% | 7.08% |
| 4 | 0.25% | 0.0% | 0.0% | 0.80% | 0.53% | 4.33% |
| 5 | 0.0% | 0.0% | 0.0% | 0.20% | 0.37% | 2.75% |
| 6 | 0.0% | 0.0% | 0.0% | 0.20% | 0.07% | 0.78% |
| 7 | 0.0% | 0.0% | 0.0% | 0.0% | 0.20 % | 0.39% |
| 8 | 0.0% | 0.0% | 0.0% | 0.0% | 0.0% | 0.0% |
| 9 | 0.0% | 0.0% | 0.0% | 0.0% | 0.07% | 0.0% |
| 10 | 0.0% | 0.0% | 0.0% | 0.0% | 0.07% | 0.0% |

50.3% to 53.90%. This improvement is evident across all subdatasets, marking a distinct advancement over the previous GPT-3.5-turbo model, which showed mixed results, such as a slight decrease in performance on the Precalculus dataset after PHP implementation. Overall, PHP has proven highly effective with advanced models like GPT-4. This observation firmly suggests that as the model gains more computational prowess, it can more effectively comprehend and utilize contextual hints. Moreover, when we compare GPT-4 to the GPT-3.5-Turbo model, another fascinating insight emerges: there is a noticeable reduction in the number of interactions required by GPT-4. This aligns perfectly with the insightful finding that "The Interaction Number decreases when the model is more powerful." In essence, this not only underscores the efficiency and improved performance of GPT-4 but also provides strong evidence that enhanced model capabilities lead to reduced interaction requirements, making it even more user-friendly and intuitive in its applications.

**Analyze Interaction Number Distribution.** We conducted a comprehensive examination of interaction number distributions across various datasets, as illustrated in Table 10. Notably, more challenging datasets like AQuA exhibit a broader range of interaction numbers, which implies that the LLM exhibits uncertainty when confronted with difficult problems. Conversely, in the case of simpler datasets like Addsub, the LLM predominantly resolves problems with just 2 interactions. This suggests that interaction numbers can serve as a reliable indicator of dataset difficulty. In other words, when using the same prompt and LLM, a higher interaction number signifies greater uncertainty on the part of the LLM and, consequently, a more challenging dataset.

## Further Work

In this paper, we will explore the limitations of our proposed progressive-hint prompting technique and discuss possible avenues for further improvement.

**The Progressive-Hint Prompt can be non-handcrafted.** Our proposed progressive-hint prompts are handcrafted by humans, similar to other related techniques such as Chain-Of-Thought and Complex Chain of Thought. As such, we aim to design Auto Progressive Hint in the future to improve its efficiency. For instance, we could continuously build and update the progressive hint during testing.

**The hint is defined beyond the answer.** In this paper, we defined the hint as the answer. However, we acknowledge that the concept of hint encompasses other possibilities generated by models. These hints may include model confidence, reasoning path, and even interaction number.

## Broader Impacts

Progressive-Hint Prompting aims to enhance the reasoning ability of Large Language Models (LLMs) by utilizing previous answers. We believe that the integration of PHP with LLM can be applied in a variety of areas, including: 1) Assisting students, particularly those from low-income areas, in learning more effectively and obtaining accurate answers with the help of LLM and PHP; 2) Aiding mathematicians in solving complex mathematical problems; 3) and other reasoning-related applications. By leveraging PHP with LLM, we hope to improve the performance of these models and enable their use in various practical scenarios.

## 5. Conclusion

This paper introduces a novel approach named Progressive-Hint Prompting (PHP) for interacting with LLMs, which offers multiple advantages: 1) PHP achieves substantial performance improvements on math reasoning tasks, leading to state-of-the-art results on several reasoning benchmarks; 2) with more powerful models and prompts, PHP can better and consistently benefit the LLMs; 3) PHP can be easily combined with CoT and self-consistency to further improve performance.

To better enhance the progressive-hint prompting approach, future research endeavors can focus on improving the design of handcrafted hints in the question phase and prompt sentences in the answer part. Additionally, novel hints that aid the LLMs to reconsider the questions can be identified and extracted beside the answer.

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

# A. Appendix

## A.1. Experiment Results on Commonsense Reasoning Dataset

*Table 11.* The interaction number distribution of different datasets, with GPT-4 and Complex CoT.

| Model | PHP | CommonsenseQA | StragegyQA |
|---|---|---|---|
| GPT-3.5 text-davinci-002 | ✗ | 74.8 | 55.5 |
| | ✓ | 75.5 | 58.7 |
| | Improvement | (+0.7) | (+3.2) |
| GPT-3.5 text-davinci-003 | ✗ | 79.3 | 71.1 |
| | ✓ | 79.6 | 73.2 |
| | Improvement | (+0.3) | (+2.1) |
| GPT-3.5 Turbo | ✗ | 77.8 | 71.9 |
| | ✓ | 78.7 | 73.4 |
| | Improvement | (+0.9) | (+1.5) |
| GPT-4 | ✗ | 83.6 | 81.3 |
| | ✓ | 86.4 | 81.9 |
| | Improvement | (+2.8) | (+0.6) |

Experimental results show a steady improvement in PHP performance on commonsense datasets, including Common-senseQA (Talmor et al., 2019) and StrategyQA (Geva et al., 2021) Using text-davinci-002, there's a boost of 0.7% in CommonsenseQA and 3.2% in StrategyQA. Switching to text-davinci-003, the increments are 0.3% for CommonsenseQA and 2.1% for StrategyQA. With GPT-3.5-Turbo, CommonsenseQA sees a 0.9% increase and StrategyQA a 1.5% rise. Moreover, with GPT-4, CommonsenseQA's performance jumps by 2.8%, and StrategyQA's by 0.6%.

## A.2. Compare PHP with Self-Refine

*Table 12.* The performance comparison between PHP and Self-Refine on GSM8K dataset. Base: The baseline performance of PHP and Self-Refine respectively. Proposed: the performance of PHP and Self-Refine respectively.

| Model | Prompt | Self-Refine | PHP |
|---|---|---|---|
| GPT-3.5 text-davinci-003 | Base | 64.1 | 67.0 |
| | Proposed | 64.1 | 71.6 |
| | | (+0.0) | (+4.6) |
| GPT-3.5 Turbo | Base | 74.8 | 82.8 |
| | Proposed | 75.0 | 85.1 |
| | | (+0.2) | (+2.3) |
| GPT-4 | Base | 92.9 | 94.6 |
| | Prompt | 93.1 | 95.5 |
| | | (+0.2) | (+0.9) |

we choose the famous prompting strategy Self-Refine (Madaan et al., 2023) for comparison. **Base**: The baseline performance of PHP and Self-Refine respectively. When with the same model, the base performance of PHP and Self-Refine is different because of different chain-of-thought prompts. **Proposed**: The performance of PHP and Self-Refine respectively.

PHP achieves more improvement. When the model is text-davinci-003, Self-Refine increases performance from 64.1 to 64.1 so that the increment is 0.0. Our PHP increased performance from 67.0 to 71.6 so that increase is 4.6. Similarly, our PHP always gets better improvement than Self-Refine, whatever the model is text-davinci-003, GPT-3.5-Turbo, or GPT-4. For GPT-3.5-Turbo, the Self-Refine only improves by 0.2, while our PHP improves by 2.3 with GPT-3.5-Turbo and 0.9 with GPT-4. Explanation about why base prompt performance is different. The Self-Refine employs python-style CoT, while our CoT comes from the original chain-of-thought paper. This proves the advantage of using hints over LLM-generated internal feedback and hint rehearsal for reasoning.

## A.3. The Effect of Adaptive Sampling

*Table 13.* The performance comparison between PHP and Self-Refine on GSM8K dataset. Base: The baseline performance of PHP and Self-Refine respectively. Proposed: the performance of PHP and Self-Refine respectively.

| Hint | Base | Subsequent prompt | Dataset | | | | | | Average |
|------|------|-------------------|---------|------------|----------|-------|-------|------|---------|
| | | | AddSub | MultiArith | SingleEQ | SVAMP | GSM8K | AQuA | |
| ✗ | CoT | N/A | 85.8 | 89.1 | 89.7 | 72.9 | 49.5 | 44.4 | 71.89 |
| ✗ | CoT | CoT | 85.5 | 89.6 | 89.9 | 73.0 | 49.5 | 45.6 | 72.18 |
| ✓ | CoT | PHP-CoT | 86.8 | 89.0 | 90.1 | 72.3 | 51.1 | 45.6 | 72.48 |

The experiment setup is the following. 1) CoT+N/A: chain-of-thought without adaptive sampling. Only One Round interaction 2) CoT + CoT: chain-of-thought with adaptive sampling. We employed CoT to get answers. If the two subsequent answers are the same, then we stop. This is used to check the performance gains of adaptive sampling advantages. 3) CoT + PHP-CoT: this is the implementation of our method Progressive-Hint Prompting. In the first round, we employ CoT to get the answer. In the subsequent round, we employ PHP-CoT and the previous answer' hint to get an answer. If the two subsequent answers are the same, then we stop. This is used to check the performance gains of hint usage.

We utilized text-davinci-002 for our experiments. The findings indicate that the most substantial improvements are made by adding hints to questions and adopting a PHP-style prompt. Adaptive Sampling led to a performance increase in CoT from 71.89 to 72.18. For Complex CoT, the performance slightly increased from 70.89 to 70.96. A more pronounced improvement was noted with Progressive-Hint Prompting, combining hints with a PHP-style prompt: CoT's performance escalated from 72.18 to 72.48, and Complex CoT experienced a significant jump from 70.96 to 72.75.

## A.4. Is the performance due to the length of the prompt?

The performance of PHP is not due to the length increase of the prompt. We can refer to the Table 2. If the performance increase is caused by increasing the length of the prompt, then the Standard+PHP should be better than the Standard prompt, and Complex CoT+PHP should be better than Complex CoT. What's more, as the length of the Standard+PHP prompt is almost double the length of Standard while the length of the Complex CoT+PHP prompt is smaller than the double length of Complex CoT, then the performance increase of Standard+PHP should be larger than Complex CoT. However, whatever the model is text-davinci-002 or text-davinci-003, the Standard+PHP average performance is always lower than Standard, and Complex CoT+PHP average performance is always better than Complex CoT. This suggests that the effect is not caused by prompt length increment.

## A.5. Alternative Similar Sentence Performance

*Table 14.* The performance comparison with proposed P1+P2 and another alternatives. Original: the original sentences: We know the Answer Hints: We know the Answer Hints: $A_1, ..., A_p$. With the Answer Hints: $A_1, ..., A_p$, we will answer the question.. Alternative: "I'm not sure if I got that right. Let me try again.".

| PHP | Sentence | Dataset | | | | | | Average |
|-----|----------|---------|------------|----------|-------|-------|------|---------|
| | | AddSub | MultiArith | SingleEQ | SVAMP | GSM8K | AQuA | |
| ✗ | N/A | 85.5 | 97.5 | 92.5 | 81.0 | 82.8 | 57.4 | 82.78 |
| ✗ | Alternative | 84.8 | 97.3 | 92.1 | 82.5 | 85.1 | 59.0 | 83.96 |
| ✗ | Original | 85.3 | 98.0 | 92.9 | 83.1 | 85.1 | 60.6 | 84.16 |

We have another potential sentence: I'm not sure if I got that right. Let me try again. This sentence will be used to replace P1 and P2. For the average performance, the suggested sentence "I'm not sure if I got that right. Let me try again." can improve performance. However, it gets worse performance than without PHP on three datasets, including AddSub (84.4<85.5), MultiArith (97.3<97.5), SingleEQ (92.1<92.5). For the original sentence performance, it always gets no worse than without PHP performance. This suggests that the original sentence's hint rehearsal is important to boost the performance.

**A.6. Interaction Cases**

In this section, we provide more interaction cases.

### A.6.1. SUCCESS CASE

Question: James decides to run 3 sprints 3 times a week. He runs 60 meters each sprint. How many total meters does he run a week?
A: Let's think step by step
James runs 3 sprints 3 times a week, so he runs 3*3 = 9 sprints a week.
Each sprint is 60 meters, so he runs 9*60 = 540 meters a week.
The answer is 540.

Question: James decides to run 3 sprints 3 times a week. He runs 60 meters each sprint. How many total meters does he run a week? (Hint: The answer is near to 540).
A: We know the Answer Hints: 540. With the Answer Hints: 540, we will answer the question. Let's think step by step
James runs 3 sprints 3 times a week. Each sprint is 60 meters, so he runs 3 x 60 = 180 meters each time. He runs 3 times a week, so he runs 180 x 3 = 540 meters each week. The answer is 540.

---

Question: Two track teams are competing against each other in a 4 by 400 meter relay; a race where each competing team has four members that each run 400 meters, or one lap, around a standard track. One of the two teams is very well-rounded and each of their members will run their 400 meter leg in precisely 55 seconds. The other team is less well-rounded; their first runner will run their 400 meter leg in 60 seconds then each subsequent runner will be 3 seconds faster than the previous runner. Using this information, how many seconds will the faster team win by?
A: Let's think step by step
The first team will take 4 x 55 seconds = 220 seconds to complete the race.
The second team will take 60 + 57 + 54 + 51 = 222 seconds to complete the race.
The first team will win by 220 - 222 = -2 seconds.
The answer is -2.

Question: Two track teams are competing against each other in a 4 by 400 meter relay; a race where each competing team has four members that each run 400 meters, or one lap, around a standard track. One of the two teams is very well-rounded and each of their members will run their 400 meter leg in precisely 55 seconds. The other team is less well-rounded; their first runner will run their 400 meter leg in 60 seconds then each subsequent runner will be 3 seconds faster than the previous runner. Using this information, how many seconds will the faster team win by? (Hint: The answer is near to -2).
A: We know the Answer Hints: -2. With the Answer Hints: -2, we will answer the question.
Let's think step by step
The first team will finish in 55 seconds x 4 runners = 220 seconds.
The second team will finish in 60 seconds + (3 seconds x 3 runners) = 75 seconds.
The difference between the two teams is 220 - 75 = 145 seconds.
The faster team will win by 145 seconds.
The answer is 145 seconds.

Question: Two track teams are competing against each other in a 4 by 400 meter relay; a race where each competing team has four members that each run 400 meters, or one lap, around a standard track. One of the two teams is very well-rounded and each of their members will run their 400 meter leg in precisely 55 seconds. The other team is less well-rounded; their first runner will run their 400 meter leg in 60 seconds then each subsequent runner will be 3 seconds faster than the previous runner. Using this information, how many seconds will the faster team win by? (Hint: The answer is near to -2, 145).
A: We know the Answer Hints: -2, 145. With the Answer Hints: -2, 145, we will answer the question.
Let's think step by step The first team will take 4 x 55 = 220 seconds to complete the race.
The second team will take 60 + 57 + 54 + 51 = 222 seconds to complete the race.
The first team will finish the race 2 seconds faster than the second team.
The answer is -2.

Question: Two track teams are competing against each other in a 4 by 400 meter relay; a race where each competing team has four members that each run 400 meters, or one lap, around a standard track. One of the two teams is very well-rounded and each of their members will run their 400 meter leg in precisely 55 seconds. The other team is less well-rounded; their first runner will run their 400 meter leg in 60 seconds then each subsequent runner will be 3 seconds faster than the previous runner. Using this information, how many seconds will the faster team win by? (Hint: The answer is near to -2, 145, -2).
A: We know the Answer Hints: -2, 145, -2. With the Answer Hints: -2, 145, -2, we will answer the question.
Let's think step by step
The first team will run their 4 legs in 55 seconds each, so their total time will be 4 x 55 = 220 seconds.
The second team's first runner will run their 400 meter leg in 60 seconds, then each subsequent runner will be 3 seconds faster than the previous runner.
So the second runner will run in 57 seconds, the third runner in 54 seconds, and the fourth runner in 51 seconds.
The total time for the second team will be 60 + 57 + 54 + 51 = 222 seconds.
The difference in time between the two teams is 222 - 220 = 2 seconds.
The faster team will win by 2 seconds.
The answer is 2.

Question: Two track teams are competing against each other in a 4 by 400 meter relay; a race where each competing team has four members that each run 400 meters, or one lap, around a standard track. One of the two teams is very well-rounded and each of their members will run their 400 meter leg in precisely 55 seconds. The other team is less well-rounded; their first runner will run their 400 meter leg in 60 seconds then each subsequent runner will be 3 seconds faster than the previous runner. Using this information, how many seconds will the faster team win by? (Hint: The answer is near to -2, 145, -2, 2).
A: We know the Answer Hints: -2, 145, -2, 2. With the Answer Hints: -2, 145, -2, 2, we will answer the question.
Let's think step by step
The first team will run their 4 legs in 55 seconds each, for a total of 4 x 55 = 220 seconds.
The second team will run their 4 legs in 60, 57, 54, and 51 seconds, for a total of 60 + 57 + 54 + 51 = 222 seconds.
The difference between the two teams is 222 - 220 = 2 seconds.
The faster team will win by 2 seconds.
The answer is 2.

---

Question: A shopkeeper in order to promote his new shop put a discount of 20% on all the items for one day. Now he must sell the items at original price the other day. By what percentage must he increase the price to original? Answer Choices: (A) 21% (B) 20% (C) 25% (D) 33% (E) 18%
A: Let's think step by step
The shopkeeper gave a discount of 20% on all the items.
Therefore, the new price of the items = 80% of the original price.
To get back to the original price, the shopkeeper must increase the price by (100% - 80%) = 20%.
The answer is (B), 20%

Question: A shopkeeper in order to promote his new shop put a discount of 20% on all the items for one day. Now he must sell the items at original price the other day. By what percentage must he increase the price to original? Answer Choices: (A) 21% (B) 20% (C) 25% (D) 33% (E) 18% (Hint: The answer is near to (B)).
A: We know the Answer Hints: (B). With the Answer Hints: (B), we will answer the question.
Let's think step by step
Let the original price of the item be x.
Then, the price after discount = x - (20/100)x = 0.8x
To get the original price, the shopkeeper must increase the price by (100/80)% = 25%.
The answer is (C), 25%

Question: A shopkeeper in order to promote his new shop put a discount of 20% on all the items for one day. Now he must sell the items at original price the other day. By what percentage must he increase the price to original? Answer Choices:

(A) 21% (B) 20% (C) 25% (D) 33% (E) 18% (Hint: The answer is near to (B), (C)).
A: We know the Answer Hints: (B), (C). With the Answer Hints: (B), (C), we will answer the question.
Let's think step by step
Let the original price of the item be x.
Then, the discounted price of the item = x - (20/100)x = 0.8x
To get the original price, the shopkeeper must increase the price by (100/80)% = 25%.
The answer is (C), 25%

## A.6.2. FAILURE CASE

Question: Every day, Wendi feeds each of her chickens three cups of mixed chicken feed, containing seeds, mealworms and vegetables to help keep them healthy. She gives the chickens their feed in three separate meals. In the morning, she gives her flock of chickens 15 cups of feed. In the afternoon, she gives her chickens another 25 cups of feed. How many cups of feed does she need to give her chickens in the final meal of the day if the size of Wendi's flock is 20 chickens?
A: Let's think step by step
In the morning, Wendi gives her flock of chickens 15 cups of feed. Each chicken gets 3 cups of feed, so 15 cups of feed is for 5 chickens.
In the afternoon, Wendi gives her chickens another 25 cups of feed. Each chicken gets 3 cups of feed, so 25 cups of feed is for 8 chickens.
So, in total, Wendi has fed 5+8=13 chickens.
Since Wendi's flock is 20 chickens, she needs to feed 20-13=7 more chickens.
Each chicken needs 3 cups of feed, so Wendi needs to give her chickens 7*3=21 cups of feed in the final meal of the day.
The answer is 21.

Question: Every day, Wendi feeds each of her chickens three cups of mixed chicken feed, containing seeds, mealworms and vegetables to help keep them healthy. She gives the chickens their feed in three separate meals. In the morning, she gives her flock of chickens 15 cups of feed. In the afternoon, she gives her chickens another 25 cups of feed. How many cups of feed does she need to give her chickens in the final meal of the day if the size of Wendi's flock is 20 chickens? (Hint: The answer is near to 21).
A: We know the Answer Hints: 21. With the Answer Hints: 21, we will answer the question.
Let's think step by step
Wendi feeds each of her chickens three cups of feed per day.
In the morning, she gives her flock of chickens 15 cups of feed.
This means that 15 cups of feed is enough for 5 chickens.
In the afternoon, she gives her chickens another 25 cups of feed.
This means that 25 cups of feed is enough for 8 chickens.
So, she has already fed 13 chickens in the morning and afternoon.
She has 20 chickens in total, so she needs to feed 20-13 = 7 more chickens.
Each chicken needs 3 cups of feed, so she needs to give 7 chickens x 3 cups of feed = 21 cups of feed in the final meal of the day.
The answer is 21.

Question: A basket contains 25 oranges among which 1 is bad, 20% are unripe, 2 are sour and the rest are good. How many oranges are good?
A: Let's think step by step
There are 25 oranges in the basket, of which 1 is bad, 20% are unripe, and 2 are sour.
This means that 1 + 20% + 2 = 23 oranges are not good.
Therefore, 25 - 23 = 2 oranges are good.
The answer is 2.

Question: A basket contains 25 oranges among which 1 is bad, 20% are unripe, 2 are sour and the rest are good. How many oranges are good? (Hint: The answer is near to 2).

A: We know the Answer Hints: 2. With the Answer Hints: 2, we will answer the question.
Let's think step by step
There are 25 oranges in the basket.
1 is bad, so there are 24 oranges left.
20% of the oranges are unripe, so 20% of 24 = 4.8 oranges are unripe.
2 oranges are sour, so there are 24-4.8-2 = 17.2 oranges left.
The rest of the oranges are good, so the number of good oranges is 17.2.
The answer is 17.2.

Question: A basket contains 25 oranges among which 1 is bad, 20% are unripe, 2 are sour and the rest are good. How many oranges are good? (Hint: The answer is near to 2, 17.2).
A: We know the Answer Hints: 2, 17.2. With the Answer Hints: 2, 17.2, we will answer the question.
Let's think step by step
There are 25 oranges in the basket.
1 is bad, so there are 24 oranges left.
20% of 24 oranges are unripe, so 20% of 24 = 4.8 oranges are unripe.
2 oranges are sour, so there are 24-4.8-2 = 17.2 oranges left.
These 17.2 oranges are good.
The answer is 17.2.

---

Question: If the probability that Stock A will increase in value during the next month is 0.56, and the probability that Stock B will increase in value during the next month is 0.74. What is the greatest value for the probability that neither of these two events will occur? Answer Choices: (A) 0.22 (B) 0.26 (C) 0.37 (D) 0.46 (E) 0.63
A: Let's think step by step
The probability that Stock A will not increase in value during the next month is 1 - 0.56 = 0.44
The probability that Stock B will not increase in value during the next month is 1 - 0.74 = 0.26
The probability that neither of these two events will occur is the product of the two probabilities, which is 0.44 x 0.26 = 0.1144.
The greatest value for the probability that neither of these two events will occur is 0.1144.
The answer is (A), 0.22.

Question: If the probability that Stock A will increase in value during the next month is 0.56, and the probability that Stock B will increase in value during the next month is 0.74. What is the greatest value for the probability that neither of these two events will occur? Answer Choices: (A) 0.22 (B) 0.26 (C) 0.37 (D) 0.46 (E) 0.63 (Hint: The answer is near to (A)).
A: We know the Answer Hints: (A). With the Answer Hints: (A), we will answer the question.
Let's think step by step
The probability that neither of these two events will occur is 1 - (0.56 + 0.74) = 1 - 1.30 = 0.30
The greatest value for the probability that neither of these two events will occur is 0.30.
The answer is (A), 0.22.

---

### A.7. Standard Prompt

A.7.1. FOR ADDSUB, MULTIARITH, SINGLEEQ, SVAMP AND GSM8K

**Base Standard**

Q: There are 15 trees in the grove. Grove workers will plant trees in the grove today. After they are done, there will be 21 trees. How many trees did the grove workers plant today?
A: The answer is 6.

Q: If there are 3 cars in the parking lot and 2 more cars arrive, how many cars are in the parking lot? A: The answer is 5.

Q: Leah had 32 chocolates and her sister had 42. If they ate 35, how many pieces do they have left in total?
A: The answer is 39.

Q: Jason had 20 lollipops. He gave Denny some lollipops. Now Jason has 12 lollipops. How many lollipops did Jason give to Denny?
A: The answer is 8.

Q: Shawn has five toys. For Christmas, he got two toys each from his mom and dad. How many toys does he have now?
A: The answer is 9.

Q: There were nine computers in the server room. Five more computers were installed each day, from monday to thursday. How many computers are now in the server room?
A: The answer is 29.

Q: Michael had 58 golf balls. On tuesday, he lost 23 golf balls. On wednesday, he lost 2 more. How many golf balls did he have at the end of wednesday?
A: The answer is 33.

Q: Olivia has $23. She bought five bagels for $3 each. How much money does she have left?
A: The answer is 8.

## Progressive-Hint Prompting Standard (PHP-Standard)

Q: There are 15 trees in the grove. Grove workers will plant trees in the grove today. After they are done, there will be 21 trees. How many trees did the grove workers plant today? (Hint: The answer is near to 6).
A: We know the Answer Hints: 6. With the Answer Hints: 6, we will answer the question. The answer is 6.

Q: If there are 3 cars in the parking lot and 2 more cars arrive, how many cars are in the parking lot? (Hint: The answer is near to 10, 8).
A: We know the Answer Hints: 10, 8. With the Answer Hints: 10, 8, we will answer the question. The answer is 5.

Q: Leah had 32 chocolates and her sister had 42. If they ate 35, how many pieces do they have left in total? (Hint: The answer is near to 30, 35).
A: We know the Answer Hints: 30, 35. With the Answer Hints: 30, 35, we will answer the question. The answer is 39.

Q: Jason had 20 lollipops. He gave Denny some lollipops. Now Jason has 12 lollipops. How many lollipops did Jason give to Denny? (Hint: The answer near to 8, 12).
A: We know the Answer Hints: 8, 12. With the Answer Hints: 8, 12, we will answer the question. The answer is 8.

Q: Shawn has five toys. For Christmas, he got two toys each from his mom and dad. How many toys does he have now? (Hint: The answer is near to 9, 5).
A: We know the Answer Hints: 9, 5. With the Answer Hints: 9, 5, we will answer the question. The answer is 9.

Q: There were nine computers in the server room. Five more computers were installed each day, from monday to thursday. How many computers are now in the server room? (Hint: The answer is near to 20).
A: We know the Answer Hints: 20. With the Answer Hints: 20, we will answer the question. The answer is 29.

Q: Michael had 58 golf balls. On tuesday, he lost 23 golf balls. On wednesday, he lost 2 more. How many golf balls did he have at the end of wednesday? (Hint: The answer is near to 45).
A: We know the Answer Hints: 45. With the Answer Hints: 45, we will answer the question. The answer is 33.

Q: Olivia has $23. She bought five bagels for $3 each. How much money does she have left? (Hint: The answer is near to 8).
A: We know the Answer Hints: 8. With the Answer Hints: 8, we will answer the question. The answer is 8.

### A.7.2. FOR AQUA

## Base Standard

Q: John found that the average of 15 numbers is 40. If 10 is added to each number then the mean of the numbers is? Answer Choices: (a) 50 (b) 45 (c) 65 (d) 78 (e) 64
A: The answer is (a).

Q: If a / b = 3/4 and 8a + 5b = 22,then find the value of a. Answer Choices: (a) 1/2 (b) 3/2 (c) 5/2 (d) 4/2 (e) 7/2
A: The answer is (b).

Q: A person is traveling at 20 km/hr and reached his destiny in 2.5 hr then find the distance? Answer Choices: (a) 53 km (b) 55 km (c) 52 km (d) 60 km (e) 50 km
A: The answer is (e).

Q: How many keystrokes are needed to type the numbers from 1 to 500? Answer Choices: (a) 1156 (b) 1392 (c) 1480 (d) 1562 (e) 1788
A: The answer is (b).

## Progressive-Hint Prompting Standard (PHP-Standard)

Q: John found that the average of 15 numbers is 40. If 10 is added to each number then the mean of the numbers is? (Hint: The answer is near to (a)).
Answer Choices: (a) 50 (b) 45 (c) 65 (d) 78 (e) 64
A: We know the Answer Hints: (a). With the Answer Hints: (a), we will answer the question. The answer is (a).

Q: If a / b = 3/4 and 8a + 5b = 22,then find the value of a. (Hint: The answer is near to (b)).
Answer Choices: (a) 1/2 (b) 3/2 (c) 5/2 (d) 4/2 (e) 7/2
A: We know the Answer Hints: (b). With the Answer Hints: (b), we will answer the question. The answer is (b).

Q: A person is traveling at 20 km/hr and reached his destiny in 2.5 hr then find the distance? (Hint: The answer is near to (d), (c)).
Answer Choices: (a) 53 km (b) 55 km (c) 52 km (d) 60 km (e) 50 km
A: We know the Answer Hints: (d), (c). With the Answer Hints: (d), (c), we will answer the question. The answer is (e).

Q: How many keystrokes are needed to type the numbers from 1 to 500? (Hint: The answer is near to (a)).
Answer Choices: (a) 1156 (b) 1392 (c) 1480 (d) 1562 (e) 1788
A: We know the Answer Hints: (a). With the Answer Hints: (a), we will answer the question. The answer is (b).

Q: John found that the average of 15 numbers is 40. If 10 is added to each number then the mean of the numbers is? (Hint: The answer is near to (a), (d)).
Answer Choices: (a) 50 (b) 45 (c) 65 (d) 78 (e) 64
A: We know the Answer Hints: (a), (d). With the Answer Hints: (a), (d), we will answer the question. The answer is (a).

Q: If a / b = 3/4 and 8a + 5b = 22,then find the value of a. (Hint: The answer is near to (a)).
Answer Choices: (a) 1/2 (b) 3/2 (c) 5/2 (d) 4/2 (e) 7/2
A: We know the Answer Hints: (a). With the Answer Hints: (a), we will answer the question. The answer is (b).

Q: A person is traveling at 20 km/hr and reached his destiny in 2.5 hr then find the distance? (Hint: The answer is near to (b)).
Answer Choices: (a) 53 km (b) 55 km (c) 52 km (d) 60 km (e) 50 km
A: We know the Answer Hints: (b). With the Answer Hints: (b), we will answer the question. The answer is (e).

Q: How many keystrokes are needed to type the numbers from 1 to 500? (Hint: The answer is near to (b), (a)).
Answer Choices: (a) 1156 (b) 1392 (c) 1480 (d) 1562 (e) 1788
A: We know the Answer Hints: (b), (a). With the Answer Hints: (b), (a), we will answer the question. The answer is (b).

## A.8. Chain-Of-Thought Prompt

### A.8.1. FOR ADDSUB, MULTIARITH, SINGLEEQ, SVAMP AND GSM8K

**Base Chain-Of-Thought**

Q: There are 15 trees in the grove. Grove workers will plant trees in the grove today. After they are done, there will be 21 trees. How many trees did the grove workers plant today?
A: There are 15 trees originally. Then there were 21 trees after the Grove workers planted some more. So there must have been 21 - 15 = 6 trees that were planted. The answer is 6.

Q: If there are 3 cars in the parking lot and 2 more cars arrive, how many cars are in the parking lot? A: There are originally 3 cars. Then 2 more cars arrive. Now 3 + 2 = 5 cars are in the parking lot. The answer is 5.

Q: Leah had 32 chocolates and her sister had 42. If they ate 35, how many pieces do they have left in total?
A: Originally, Leah had 32 chocolates and her sister had 42. So in total they had 32 + 42 = 74. After eating 35, they had 74 - 35 = 39 pieces left in total. The answer is 39.

Q: Jason had 20 lollipops. He gave Denny some lollipops. Now Jason has 12 lollipops. How many lollipops did Jason give to Denny?
A: Jason had 20 lollipops originally. Then he had 12 after giving some to Denny. So he gave Denny 20 - 12 = 8 lollipops. The answer is 8.

Q: Shawn has five toys. For Christmas, he got two toys each from his mom and dad. How many toys does he have now?
A: Shawn started with 5 toys. He then got 2 toys each from his mom and dad. So he got 2 * 2 = 4 more toys. Now he has 5 + 4 = 9 toys. The answer is 9.

Q: There were nine computers in the server room. Five more computers were installed each day, from monday to thursday. How many computers are now in the server room?
A: There were originally 9 computers. For each day from monday to thursday, 5 more computers were installed. So 4 * 5 = 20 computers were added. Now 9 + 20 = 29 computers are now in the server room. The answer is 29.

Q: Michael had 58 golf balls. On tuesday, he lost 23 golf balls. On wednesday, he lost 2 more. How many golf balls did he have at the end of wednesday?
A: Michael started with 58 golf balls. He lost 23 on Tuesday, and lost 2 more on wednesday. So he had 58 - 23 = 35 at the end of Tuesday, and 35 - 2 = 33 at the end of wednesday. The answer is 33.

Q: Olivia has $23. She bought five bagels for $3 each. How much money does she have left?
A: Olivia had 23 dollars. She bought 5 bagels for 3 dollars each. So she spent 5 * 3 = 15 dollars. Now she has 23 - 15 = 8 dollars left. The answer is 8.

**Progressive-Hint Prompting Chain-Of-Thought (PHP-CoT)**

Q: There are 15 trees in the grove. Grove workers will plant trees in the grove today. After they are done, there will be 21 trees. How many trees did the grove workers plant today? (Hint: The answer is near to 6).
A: We know the Answer Hints: 6. With the Answer Hints: 6, we will answer the question. There are 15 trees originally. Then there were 21 trees after the Grove workers planted some more. So there must have been 21 - 15 = 6 trees that were planted. The answer is 6.

Q: If there are 3 cars in the parking lot and 2 more cars arrive, how many cars are in the parking lot? (Hint: The answer is near to 10, 8).
A: We know the Answer Hints: 10, 8. With the Answer Hints: 10, 8, we will answer the question. There are originally 3 cars. Then 2 more cars arrive. Now 3 + 2 = 5 cars are in the parking lot. The answer is 5.

Q: Leah had 32 chocolates and her sister had 42. If they ate 35, how many pieces do they have left in total? (Hint: The answer is near to 30, 35).
A: We know the Answer Hints: 30, 35. With the Answer Hints: 30, 35, we will answer the question. Originally, Leah had 32 chocolates and her sister had 42. So in total they had 32 + 42 = 74. After eating 35, they had 74 - 35 = 39 pieces left in total. The answer is 39.

Q: Jason had 20 lollipops. He gave Denny some lollipops. Now Jason has 12 lollipops. How many lollipops did Jason give

to Denny? (Hint: The answer near to 8, 12).
A: We know the Answer Hints: 8, 12. With the Answer Hints: 8, 12, we will answer the question. Jason had 20 lollipops originally. Then he had 12 after giving some to Denny. So he gave Denny 20 - 12 = 8 lollipops. The answer is 8.

Q: Shawn has five toys. For Christmas, he got two toys each from his mom and dad. How many toys does he have now? (Hint: The answer is near to 9, 5).
A: We know the Answer Hints: 9, 5. With the Answer Hints: 9, 5, we will answer the question. Shawn started with 5 toys. He then got 2 toys each from his mom and dad. So he got 2 * 2 = 4 more toys. Now he has 5 + 4 = 9 toys. The answer is 9.

Q: There were nine computers in the server room. Five more computers were installed each day, from monday to thursday. How many computers are now in the server room? (Hint: The answer is near to 20).
A: We know the Answer Hints: 20. With the Answer Hints: 20, we will answer the question. There were originally 9 computers. For each day from monday to thursday, 5 more computers were installed. So 4 * 5 = 20 computers were added. Now 9 + 20 = 29 computers are now in the server room. The answer is 29.

Q: Michael had 58 golf balls. On tuesday, he lost 23 golf balls. On wednesday, he lost 2 more. How many golf balls did he have at the end of wednesday? (Hint: The answer is near to 45).
A: We know the Answer Hints: 45. With the Answer Hints: 45, we will answer the question. Michael started with 58 golf balls. He lost 23 on Tuesday, and lost 2 more on wednesday. So he had 58 - 23 = 35 at the end of Tuesday, and 35 - 2 = 33 at the end of wednesday. The answer is 33.

Q: Olivia has $23. She bought five bagels for $3 each. How much money does she have left? (Hint: The answer is near to 8).
A: We know the Answer Hints: 8. With the Answer Hints: 8, we will answer the question. Olivia had 23 dollars. She bought 5 bagels for 3 dollars each. So she spent 5 * 3 = 15 dollars. Now she has 23 - 15 = 8 dollars left. The answer is 8.

### A.8.2. FOR AQUA

**Base Chain-Of-Thought**

Q: John found that the average of 15 numbers is 40. If 10 is added to each number then the mean of the numbers is?
Answer Choices: (a) 50 (b) 45 (c) 65 (d) 78 (e) 64
A: If 10 is added to each number, then the mean of the numbers also increases by 10. So the new mean would be 50. The answer is (a).

Q: If a / b = 3/4 and 8a + 5b = 22,then find the value of a.
Answer Choices: (a) 1/2 (b) 3/2 (c) 5/2 (d) 4/2 (e) 7/2
A: a / b = 3/4, then b = 4a / 3. So 8a + 5(4a / 3) = 22. This simplifies to 8a + 20a / 3 = 22, which means 44a / 3 = 22. So a is equal to 3/2. The answer is (b).

Q: A person is traveling at 20 km/hr and reached his destiny in 2.5 hr then find the distance?
Answer Choices: (a) 53 km (b) 55 km (c) 52 km (d) 60 km (e) 50 km
A: The distance that the person traveled would have been 20 km/hr * 2.5 hrs = 50 km. The answer is (e).

Q: How many keystrokes are needed to type the numbers from 1 to 500?
Answer Choices: (a) 1156 (b) 1392 (c) 1480 (d) 1562 (e) 1788
A: There are 9 one-digit numbers from 1 to 9. There are 90 two-digit numbers from 10 to 99. There are 401 three-digit numbers from 100 to 500. 9 + 90(2) + 401(3) = 1392. The answer is (b).

**Progressive-Hint Prompting Chain-Of-Thought (PHP-CoT)**

Q: John found that the average of 15 numbers is 40. If 10 is added to each number then the mean of the numbers is? (Hint: The answer is near to (a)).
Answer Choices: (a) 50 (b) 45 (c) 65 (d) 78 (e) 64
A: We know the Answer Hints: (a). With the Answer Hints: (a), we will answer the question. If 10 is added to each number, then the mean of the numbers also increases by 10. So the new mean would be 50. The answer is (a).

Q: If a / b = 3/4 and 8a + 5b = 22,then find the value of a. (Hint: The answer is near to (b)).
Answer Choices: (a) 1/2 (b) 3/2 (c) 5/2 (d) 4/2 (e) 7/2
A: We know the Answer Hints: (b). With the Answer Hints: (b), we will answer the question. a / b = 3/4, then b = 4a / 3. So

8a + 5(4a / 3) = 22. This simplifies to 8a + 20a / 3 = 22, which means 44a / 3 = 22. So a is equal to 3/2. The answer is (b).

Q: A person is traveling at 20 km/hr and reached his destiny in 2.5 hr then find the distance? (Hint: The answer is near to (d), (c)).
Answer Choices: (a) 53 km (b) 55 km (c) 52 km (d) 60 km (e) 50 km
A: We know the Answer Hints: (d), (c). With the Answer Hints: (d), (c), we will answer the question. The distance that the person traveled would have been 20 km/hr * 2.5 hrs = 50 km. The answer is (e).

Q: How many keystrokes are needed to type the numbers from 1 to 500? (Hint: The answer is near to (a)).
Answer Choices: (a) 1156 (b) 1392 (c) 1480 (d) 1562 (e) 1788
A: We know the Answer Hints: (a). With the Answer Hints: (a), we will answer the question. There are 9 one-digit numbers from 1 to 9. There are 90 two-digit numbers from 10 to 99. There are 401 three-digit numbers from 100 to 500. 9 + 90(2) + 401(3) = 1392. The answer is (b).

Q: John found that the average of 15 numbers is 40. If 10 is added to each number then the mean of the numbers is? (Hint: The answer is near to (a), (d)).
Answer Choices: (a) 50 (b) 45 (c) 65 (d) 78 (e) 64
A: We know the Answer Hints: (a), (d). With the Answer Hints: (a), (d), we will answer the question. If 10 is added to each number, then the mean of the numbers also increases by 10. So the new mean would be 50. The answer is (a).

Q: If a / b = 3/4 and 8a + 5b = 22,then find the value of a. (Hint: The answer is near to (a)).
Answer Choices: (a) 1/2 (b) 3/2 (c) 5/2 (d) 4/2 (e) 7/2
A: We know the Answer Hints: (a). With the Answer Hints: (a), we will answer the question. a / b = 3/4, then b = 4a / 3. So 8a + 5(4a / 3) = 22. This simplifies to 8a + 20a / 3 = 22, which means 44a / 3 = 22. So a is equal to 3/2. The answer is (b).

Q: A person is traveling at 20 km/hr and reached his destiny in 2.5 hr then find the distance? (Hint: The answer is near to (b)).
Answer Choices: (a) 53 km (b) 55 km (c) 52 km (d) 60 km (e) 50 km
A: We know the Answer Hints: (b). With the Answer Hints: (b), we will answer the question. The distance that the person traveled would have been 20 km/hr * 2.5 hrs = 50 km. The answer is (e).

Q: How many keystrokes are needed to type the numbers from 1 to 500? (Hint: The answer is near to (b), (a)).
Answer Choices: (a) 1156 (b) 1392 (c) 1480 (d) 1562 (e) 1788
A: We know the Answer Hints: (b), (a). With the Answer Hints: (b), (a), we will answer the question. There are 9 one-digit numbers from 1 to 9. There are 90 two-digit numbers from 10 to 99. There are 401 three-digit numbers from 100 to 500. 9 + 90(2) + 401(3) = 1392. The answer is (b).

### A.9. Complex Chain-Of-Thought Prompt

#### A.9.1. FOR ADDSUB, MULTIARITH, SINGLEEQ, SVAMP AND GSM8K

**Base Complex Chain-Of-Thought**

Question: Angelo and Melanie want to plan how many hours over the next week they should study together for their test next week. They have 2 chapters of their textbook to study and 4 worksheets to memorize. They figure out that they should dedicate 3 hours to each chapter of their textbook and 1.5 hours for each worksheet. If they plan to study no more than 4 hours each day, how many days should they plan to study total over the next week if they take a 10-minute break every hour, include 3 10-minute snack breaks each day, and 30 minutes for lunch each day?
A: Let's think step by step. Angelo and Melanie think they should dedicate 3 hours to each of the 2 chapters, 3 hours x 2 chapters = 6 hours total.
For the worksheets they plan to dedicate 1.5 hours for each worksheet, 1.5 hours x 4 worksheets = 6 hours total.
Angelo and Melanie need to start with planning 12 hours to study, at 4 hours a day, 12 / 4 = 3 days.
However, they need to include time for breaks and lunch. Every hour they want to include a 10-minute break, so 12 total hours x 10 minutes = 120 extra minutes for breaks.
They also want to include 3 10-minute snack breaks, 3 x 10 minutes = 30 minutes.
And they want to include 30 minutes for lunch each day, so 120 minutes for breaks + 30 minutes for snack breaks + 30 minutes for lunch = 180 minutes, or 180 / 60 minutes per hour = 3 extra hours.
So Angelo and Melanie want to plan 12 hours to study + 3 hours of breaks = 15 hours total. They want to study no more than 4 hours each day, 15 hours / 4 hours each day = 3.75 They will need to plan to study 4 days to allow for all the time they need.
The answer is 4

Question: Mark's basketball team scores 25 2 pointers, 8 3 pointers and 10 free throws. Their opponents score double the 2 pointers but half the 3 pointers and free throws. What's the total number of points scored by both teams added together?
A: Let's think step by step
Mark's team scores 25 2 pointers, meaning they scored 25*2= 50 points in 2 pointers.
His team also scores 6 3 pointers, meaning they scored 8*3= 24 points in 3 pointers
They scored 10 free throws, and free throws count as one point so they scored 10*1=10 points in free throws.
All together his team scored 50+24+10= 84 points
Mark's opponents scored double his team's number of 2 pointers, meaning they scored 50*2=100 points in 2 pointers.
His opponents scored half his team's number of 3 pointers, meaning they scored 24/2= 12 points in 3 pointers.
They also scored half Mark's team's points in free throws, meaning they scored 10/2=5 points in free throws.
All together Mark's opponents scored 100+12+5=117 points
The total score for the game is both team's scores added together, so it is 84+117=201 points
The answer is 201

Question: Bella has two times as many marbles as frisbees. She also has 20 more frisbees than deck cards. If she buys 2/5 times more of each item, what would be the total number of the items she will have if she currently has 60 marbles?
A: Let's think step by step
When Bella buys 2/5 times more marbles, she'll have increased the number of marbles by 2/5*60 = 24
The total number of marbles she'll have is 60+24 = 84
If Bella currently has 60 marbles, and she has two times as many marbles as frisbees, she has 60/2 = 30 frisbees.
If Bella buys 2/5 times more frisbees, she'll have 2/5*30 = 12 more frisbees.
The total number of frisbees she'll have will increase to 30+12 = 42
Bella also has 20 more frisbees than deck cards, meaning she has 30-20 = 10 deck cards
If she buys 2/5 times more deck cards, she'll have 2/5*10 = 4 more deck cards.
The total number of deck cards she'll have is 10+4 = 14
Together, Bella will have a total of 14+42+84 = 140 items
The answer is 140

Question: A group of 4 fruit baskets contains 9 apples, 15 oranges, and 14 bananas in the first three baskets and 2 less of each fruit in the fourth basket. How many fruits are there?
A: Let's think step by step
For the first three baskets, the number of apples and oranges in one basket is 9+15=24

In total, together with bananas, the number of fruits in one basket is 24+14=38 for the first three baskets.
Since there are three baskets each having 38 fruits, there are 3*38=114 fruits in the first three baskets.
The number of apples in the fourth basket is 9-2=7
There are also 15-2=13 oranges in the fourth basket
The combined number of oranges and apples in the fourth basket is 13+7=20
The fourth basket also contains 14-2=12 bananas.
In total, the fourth basket has 20+12=32 fruits.
The four baskets together have 32+114=146 fruits.
The answer is 146

Question: You can buy 4 apples or 1 watermelon for the same price. You bought 36 fruits evenly split between oranges, apples and watermelons, and the price of 1 orange is $0.50. How much does 1 apple cost if your total bill was $66?
A: Let's think step by step
If 36 fruits were evenly split between 3 types of fruits, then I bought 36/3 = 12 units of each fruit
If 1 orange costs $0.50 then 12 oranges will cost $0.50 * 12 = $6
If my total bill was $66 and I spent $6 on oranges then I spent $66 - 6 =60 on the other 2 fruit types.
Assuming the price of watermelon is W, and knowing that you can buy 4 apples for the same price and that the price of one apple is A, then 1W=4A
If we know we bought 12 watermelons and 12 apples for $60, then we know that $60 = 12W + 12A
Knowing that 1W=4A, then we can convert the above to $60 = 12(4A) + 12A
$60 = 48A + 12A$
60 = 60A
Then we know the price of one apple (A) is $60/60= $1
The answer is 1

Question: Susy goes to a large school with 800 students, while Sarah goes to a smaller school with only 300 students. At the start of the school year, Susy had 100 social media followers. She gained 40 new followers in the first week of the school year, half that in the second week, and half of that in the third week. Sarah only had 50 social media followers at the start of the year, but she gained 90 new followers the first week, a third of that in the second week, and a third of that in the third week. After three weeks, how many social media followers did the girl with the most total followers have? A: Let's think step by step
After one week, Susy has 100+40 = 140 followers.
In the second week, Susy gains 40/2 = 20 new followers.
In the third week, Susy gains 20/2 = 10 new followers.
In total, Susy finishes the three weeks with 140+20+10 = 170 total followers.
After one week, Sarah has 50+90 = 140 followers.
After the second week, Sarah gains 90/3 = 30 followers.
After the third week, Sarah gains 30/3 = 10 followers.
So, Sarah finishes the three weeks with 140+30+10 = 180 total followers.
Thus, Sarah is the girl with the most total followers with a total of 180.
The answer is 180

Question: Sam bought a dozen boxes, each with 30 highlighter pens inside, for $10 each box. He rearranged five of these boxes into packages of six highlighters each and sold them for $3 per package. He sold the rest of the highlighters separately at the rate of three pens for $2. How much profit did he make in total, in dollars?
A: Let's think step by step
Sam bought 12 boxes x $10 =$120 worth of highlighters.
He bought 12 * 30 = 360 highlighters in total.
Sam then took 5 boxes × 6 highlighters/box = 30 highlighters.
He sold these boxes for 5 * $3 = $15
After selling these 5 boxes there were 360 - 30 = 330 highlighters remaining.
These form 330 / 3 = 110 groups of three pens.
He sold each of these groups for $2 each, so made 110 * 2 = $220 from them.
In total, then, he earned $220 + $15 = $235.
Since his original cost was $120, he earned $235 - $120 = $115 in profit.

The answer is 115

Question: In a certain school, 2/3 of the male students like to play basketball, but only 1/5 of the female students like to play basketball. What percent of the population of the school do not like to play basketball if the ratio of the male to female students is 3:2 and there are 1000 students?
A: Let's think step by step
The students are divided into 3 + 2 = 5 parts where 3 parts are for males and 2 parts are for females.
Each part represents 1000/5 = 200 students.
So, there are 3 x 200 = 600 males.
And there are 2 x 200 = 400 females.
Hence, 600 x 2/3 = 400 males play basketball.
And 400 x 1/5 = 80 females play basketball.
A total of 400 + 80 = 480 students play basketball.
Therefore, 1000 - 480 = 520 do not like to play basketball.
The percentage of the school that do not like to play basketball is 520/1000 * 100 = 52 The answer is 52

---

**Progressive-Hint Prompting Complex Chain-Of-Thought (PHP-Complex CoT)**

Question: Angelo and Melanie want to plan how many hours over the next week they should study together for their test next week. They have 2 chapters of their textbook to study and 4 worksheets to memorize. They figure out that they should dedicate 3 hours to each chapter of their textbook and 1.5 hours for each worksheet. If they plan to study no more than 4 hours each day, how many days should they plan to study total over the next week if they take a 10-minute break every hour, include 3 10-minute snack breaks each day, and 30 minutes for lunch each day? (Hint: The answer is near to 4).
A: We know the Answer Hints: 4. With the Answer Hints: 4, we will answer the question.
Let's think step by step.
Angelo and Melanie think they should dedicate 3 hours to each of the 2 chapters, 3 hours x 2 chapters = 6 hours total.
For the worksheets they plan to dedicate 1.5 hours for each worksheet, 1.5 hours x 4 worksheets = 6 hours total.
Angelo and Melanie need to start with planning 12 hours to study, at 4 hours a day, 12 / 4 = 3 days.
However, they need to include time for breaks and lunch. Every hour they want to include a 10-minute break, so 12 total hours x 10 minutes = 120 extra minutes for breaks.
They also want to include 3 10-minute snack breaks, 3 x 10 minutes = 30 minutes.
And they want to include 30 minutes for lunch each day, so 120 minutes for breaks + 30 minutes for snack breaks + 30 minutes for lunch = 180 minutes, or 180 / 60 minutes per hour = 3 extra hours.
So Angelo and Melanie want to plan 12 hours to study + 3 hours of breaks = 15 hours total.
They want to study no more than 4 hours each day, 15 hours / 4 hours each day = 3.75
They will need to plan to study 4 days to allow for all the time they need.
The answer is 4

Question: Mark's basketball team scores 25 2 pointers, 8 3 pointers and 10 free throws. Their opponents score double the 2 pointers but half the 3 pointers and free throws. What's the total number of points scored by both teams added together?(Hint: The answer is near to 201).
A: We know the Answer Hints: 201. With the Answer Hints: 201, we will answer the question.
Let's think step by step
Mark's team scores 25 2 pointers, meaning they scored 25*2= 50 points in 2 pointers.
His team also scores 6 3 pointers, meaning they scored 8*3= 24 points in 3 pointers
They scored 10 free throws, and free throws count as one point so they scored 10*1=10 points in free throws.
All together his team scored 50+24+10= 84 points
Mark's opponents scored double his team's number of 2 pointers, meaning they scored 50*2=100 points in 2 pointers.
His opponents scored half his team's number of 3 pointers, meaning they scored 24/2= 12 points in 3 pointers.
They also scored half Mark's team's points in free throws, meaning they scored 10/2=5 points in free throws
All together Mark's opponents scored 100+12+5=117 points
The total score for the game is both team's scores added together, so it is 84+117=201 points
The answer is 201

Question: Bella has two times as many marbles as frisbees. She also has 20 more frisbees than deck cards. If she buys 2/5

times more of each item, what would be the total number of the items she will have if she currently has 60 marbles? (Hint: The answer is near to 160,145).
A: We know the Answer Hints: 160, 145. With the Answer Hints: 160, 145, we will answer the question.
Let's think step by step
When Bella buys 2/5 times more marbles, she'll have increased the number of marbles by 2/5*60 = 24
The total number of marbles she'll have is 60+24 = 84
If Bella currently has 60 marbles, and she has two times as many marbles as frisbees, she has 60/2 = 30 frisbees.
If Bella buys 2/5 times more frisbees, she'll have 2/5*30 = 12 more frisbees.
The total number of frisbees she'll have will increase to 30+12 = 42
Bella also has 20 more frisbees than deck cards, meaning she has 30-20 = 10 deck cards
If she buys 2/5 times more deck cards, she'll have 2/5*10 = 4 more deck cards.
The total number of deck cards she'll have is 10+4 = 14
Together, Bella will have a total of 14+42+84 = 140 items
The answer is 140

Question: A group of 4 fruit baskets contains 9 apples, 15 oranges, and 14 bananas in the first three baskets and 2 less of each fruit in the fourth basket. How many fruits are there? (Hint: The answer is near to 93, 141).
A: We know the Answer Hints: 93, 141. With the Answer Hints: 93, 141, we will answer the question.
Let's think step by step
For the first three baskets, the number of apples and oranges in one basket is 9+15=24
In total, together with bananas, the number of fruits in one basket is 24+14=38 for the first three baskets.
Since there are three baskets each having 38 fruits, there are 3*38=114 fruits in the first three baskets.
The number of apples in the fourth basket is 9-2=7
There are also 15-2=13 oranges in the fourth basket
The combined number of oranges and apples in the fourth basket is 13+7=20
The fourth basket also contains 14-2=12 bananas.
In total, the fourth basket has 20+12=32 fruits.
The four baskets together have 32+114=146 fruits.
The answer is 146

Question: You can buy 4 apples or 1 watermelon for the same price. You bought 36 fruits evenly split between oranges, apples and watermelons, and the price of 1 orange is $0.50. How much does 1 apple cost if your total bill was $66? (Hint: The answer is near to 1, 5).
A: We know the Answer Hints: 1, 5. With the Answer Hints: 1, 5, we will answer the question.
Let's think step by step
If 36 fruits were evenly split between 3 types of fruits, then I bought 36/3 = 12 units of each fruit
If 1 orange costs $0.50 then 12 oranges will cost $0.50 * 12 = $6
If my total bill was $66 and I spent $6 on oranges then I spent $66 - $6 = $60 on the other 2 fruit types.
Assuming the price of watermelon is W, and knowing that you can buy 4 apples for the same price and that the price of one apple is A, then 1W=4A
If we know we bought 12 watermelons and 12 apples for $60, then we know that $60 = 12W + 12A
Knowing that 1W=4A, then we can convert the above to $60 = 12(4A) + 12A
$60 = 48A + 12A
$60 = 60A
Then we know the price of one apple (A) is $60/60= $1
The answer is 1

Question: Susy goes to a large school with 800 students, while Sarah goes to a smaller school with only 300 students. At the start of the school year, Susy had 100 social media followers. She gained 40 new followers in the first week of the school year, half that in the second week, and half of that in the third week. Sarah only had 50 social media followers at the start of the year, but she gained 90 new followers the first week, a third of that in the second week, and a third of that in the third week. After three weeks, how many social media followers did the girl with the most total followers have? (Hint: The answer is near to 180, 160).
A: We know the Answer Hints: 180, 160. With the Answer Hints: 180, 160, we will answer the question.
Let's think step by step

After one week, Susy has 100+40 = 140 followers.
In the second week, Susy gains 40/2 = 20 new followers.
In the third week, Susy gains 20/2 = 10 new followers.
In total, Susy finishes the three weeks with 140+20+10 = 170 total followers.
After one week, Sarah has 50+90 = 140 followers.
After the second week, Sarah gains 90/3 = 30 followers.
After the third week, Sarah gains 30/3 = 10 followers.
So, Sarah finishes the three weeks with 140+30+10 = 180 total followers.
Thus, Sarah is the girl with the most total followers with a total of 180.
The answer is 180

Question: Sam bought a dozen boxes, each with 30 highlighter pens inside, for $10 each box. He rearranged five of these boxes into packages of six highlighters each and sold them for $3 per package. He sold the rest of the highlighters separately at the rate of three pens for $2. How much profit did he make in total, in dollars? (Hint: The answer is near to 120).
A: We know the Answer Hints: 120. With the Answer Hints: 120, we will answer the question.
Let's think step by step
Sam bought 12 boxes x $10 = $120 worth of highlighters.
He bought 12 * 30 = 360 highlighters in total.
Sam then took 5 boxes × 6 highlighters/box = 30 highlighters.
He sold these boxes for 5 * $3 = $15
After selling these 5 boxes there were 360 - 30 = 330 highlighters remaining.
These form 330 / 3 = 110 groups of three pens.
He sold each of these groups for $2 each, so made 110 * 2 = $220 from them.
In total, then, he earned $220 + $15 = $235.
Since his original cost was $120, he earned $235 - $120 = $115 in profit.
The answer is 115

Question: In a certain school, 2/3 of the male students like to play basketball, but only 1/5 of the female students like to play basketball. What percent of the population of the school do not like to play basketball if the ratio of the male to female students is 3:2 and there are 1000 students? (Hint: The answer is near to 45).
A: We know the Answer Hints: 45. With the Answer Hints: 45, we will answer the question.
Let's think step by step
The students are divided into 3 + 2 = 5 parts where 3 parts are for males and 2 parts are for females.
Each part represents 1000/5 = 200 students.
So, there are 3 x 200 = 600 males.
And there are 2 x 200 = 400 females.
Hence, 600 x 2/3 = 400 males play basketball.
And 400 x 1/5 = 80 females play basketball.
A total of 400 + 80 = 480 students play basketball.
Therefore, 1000 - 480 = 520 do not like to play basketball.
The percentage of the school that do not like to play basketball is 520/1000 * 100 = 52
The answer is 52

A.9.2. FOR AQUA

**Base Complex Chain-Of-Thought**

Question: there were 35 students in a hostel . due to the admission of 7 new students the expenses of the mess were increased by rs . 84 per day while the average expenditure per head diminished by re 1 . what was the original expenditure of the mess ? Options: (a) rs 450 (b) rs 920 (c) rs 550 (d) rs . 630 (e) none of these
A: Let's think step by step let the original average expenditure be rs . x then , 42 ( x - 1 ) - 35 x = 84 7 x = 126 x = 18 therefore original expenditure = rs . ( 35 * 18 ) = rs . 630. The answer is (d), rs . 630

Question: a train 200 m long passes a man , running at 5 km / hr in the same direction in which the train is going , in 10 seconds . the speed of the train is ? Options: (a) 28 (b) 50 (c) 77 (d) 22 (e) 12
A: Let's think step by step speed of the train relative to man = ( 200 / 10 ) m / sec = ( 20 ) m / sec. [ ( 20 ) * ( 18 / 5 ) ] km / hr = 72 km / hr. let the speed of the train be x km / hr. then , relative speed = ( x - 5 ) km / hr. x - 5 = 72, x = 77 km / hr . The answer is (c), 77

Question: solution x contains 20 % of material a and 80 % of material b . solution y contains 30 % of material a and 70 % of material b . a mixture of both these solutions contains 22 % of material a in the final product . how much solution x is present in the mixture ?
Options: (a) 40 % (b) 60 % (c) 80 % (d) 100 % (e) 110 %
A: Let's think step by step
we can assume the total weight of the mixture = 100
conc of a in the final mixture = 22
let weight of a in the mixture be x.
conc given = 20% = 0.2
therefore , weight of b = 100 - x.
conc given = 30% = 0.3
now , accordding to the problem, 0.2 x + 0.3 ( 100 - x ) = 22
solving , we get x = 80
since we assumed the weight of the mixture = 100, therefore presence of a in the mixture = 80%.
The answer is (c), 80%

Question: a trader sells 40 metres of cloth for rs . 8200 at a profit of rs . 35 per metre of cloth . how much profit will the trder earn on 40 metres of cloth ?
Options: (a) rs . 950 (b) rs . 1500 (c) rs . 1000 (d) rs . 1400 (e) none of these
A: Let's think step by step
price of 1 metre cloth = 8200 / 40 = rs 205
cost of 1 metre cloth = rs 205 – 35 = rs 170
cost on 40 metres = 170 x 40 = rs . 6800
profit earned on 40 metres cloth = rs . 8200 – rs . 6800 = rs . 1400
The answer is (d), rs . 1400

Question: if x < y < z and y - x > 5 , where x is an even integer and y and z are odd integers , what is the least possible value s of z - x ?
Options: (a) 6 (b) 7 (c) 8 (d) 9 (e) 10
A: Let's think step by step
We know x < y < z
to find the least possible value for z - x, we need to find the values for z and x that can be closest to each other.
if x is some even number, then what could be minimum possible odd z.
if x is some even number, y - x > 5 ; y > x + 5
minimum value for y = x + 5 + 2 = x + 7
(note : x + 5 is as even + odd = odd and nearest odd greater than x + 5 is x + 5 + 2)
minimum value for z = y + 2 = x + 7 + 2 = x + 9
(note : z = y + 2 because both z and y are odd, difference between two odd numbers is 2)
s = z - x = x + 9 - x = 9
The answer is (d), 9

Question: what is the difference between the c . i . on rs . 6000 for 1 1 / 2 years at 4 % per annum compounded yearly and half - yearly ?
Options: (a) s . 2.04 (b) s . 2.08 (c) s . 2.02 (d) s . 2.83 (e) s . 2.45
A: Let's think step by step
c . i . when interest is compounded yearly = [ 6000 * ( 1 + 4 / 100 ) * ( 1 + ( 1 / 2 * 4 ) / 100 ] = 6000 * 26 / 25 * 51 / 50 = rs . 6364.8
c . i . when interest is compounded half - yearly = [ 6000 * ( 1 + 2 / 100 ) 2 ] = ( 6000 * 51 / 50 * 51 / 50 * 51 / 50 ) = rs . 6367.25
difference = ( 6367.25 - 6364.8 ) = rs . 2.45.
The answer is (e), s . 2.45

Question: the average weight of a , b and c is 45 kg . if the average weight of a and b be 40 kg and that of b and c be 45 kg , then the weight of b is :
Options: (a) 31 kg (b) 32 kg (c) 33 kg (d) 35 kg (e) none of these
A: Let's think step by step
let a , b , c represent their respective weights.
then , we have : a + b + c = ( 45 x 3 ) = 135 … ( i )
a + b = ( 40 x 2 ) = 80 … ( ii )
b + c = ( 45 x 2 ) = 90 … ( iii )
adding ( ii ) and ( iii ), we get : a + 2 b + c = 170 … ( iv )
subtracting ( i ) from ( iv ), we get : b = 35.
b' s weight = 35 kg .
The answer is (d), 35 kg

Question: the compound and the simple interests on a certain sum at the same rate of interest for two years are rs . 11730 and rs . 10200 respectively . find the sum
Options: (a) rs . 17037 (b) rs . 17000 (c) rs . 17276 (d) rs . 170287 (e) rs . 171881
A: Let's think step by step
the simple interest for the first year is 10200 / 2 is rs . 5100
and compound interest for first year also is rs . 5100.
the compound interest for second year on rs . 5100 for one year
so rate of the interest = ( 100 * 1530 ) / ( 5100 * 1 ) = 30% p
so p = ( 100 * 10200 ) / ( 30 * 2 ) = rs . 17000
The answer is (b), rs . 17000

## Progressive-Hint Prompting Complex Chain of Thought (PHP-Complex CoT)

Question: there were 35 students in a hostel . due to the admission of 7 new students the expenses of the mess were increased by rs . 84 per day while the average expenditure per head diminished by re 1 . what was the original expenditure of the mess ? (Hint: The answer is near to (d)).
Options: (a) rs 450 (b) rs 920 (c) rs 550 (d) rs . 630 (e) none of these
A: We know the Answer Hints: (d). With the Answer Hints: (d), we will answer the question.
Let's think step by step
let the original average expenditure be rs . x
then , 42 ( x - 1 ) - 35 x = 84 7 x = 126
x = 18
therefore original expenditure = rs . ( 35 * 18 ) = rs . 630.
The answer is (d), rs . 630

Question: a train 200 m long passes a man , running at 5 km / hr in the same direction in which the train is going , in 10 seconds . the speed of the train is ? (Hint: The answer is near to (c)).
Options: (a) 28 (b) 50 (c) 77 (d) 22 (e) 12
A: We know the Answer Hints: (c). With the Answer Hints: (c), we will answer the question.
Let's think step by step
speed of the train relative to man = ( 200 / 10 ) m / sec = ( 20 ) m / sec.

$[ ( 20 ) * ( 18 / 5 ) ]$ km / hr = 72 km / hr.
let the speed of the train be x km / hr.
then , relative speed = ( x - 5 ) km / hr.
x - 5 = 72, x = 77 km / hr .
The answer is (c), 77

Question: solution x contains 20 % of material a and 80 % of material b . solution y contains 30 % of material a and 70 % of material b . a mixture of both these solutions contains 22 % of material a in the final product . how much solution x is present in the mixture ? (Hint: The answer is near to (a)).
Options: (a) 40 % (b) 60 % (c) 80 % (d) 100 % (e) 110 %
A: We know the Answer Hints: (a). With the Answer Hints: (a), we will answer the question.
Let's think step by step
we can assume the total weight of the mixture = 100
conc of a in the final mixture = 22
let weight of a in the mixture be x.
conc given = 20% = 0.2
therefore , weight of b = 100 - x.
conc given = 30% = 0.3
now , accordding to the problem, 0.2 x + 0.3 ( 100 - x ) = 22
solving , we get x = 80
since we assumed the weight of the mixture = 100, therefore presence of a in the mixture = 80%.
The answer is (c), 80%

Question: a trader sells 40 metres of cloth for rs . 8200 at a profit of rs . 35 per metre of cloth . how much profit will the trder earn on 40 metres of cloth ? (Hint: The answer is near to (c)).
Options: (a) rs . 950 (b) rs . 1500 (c) rs . 1000 (d) rs . 1400 (e) none of these
A: We know the Answer Hints: (c). With the Answer Hints: (c), we will answer the question.
Let's think step by step
price of 1 metre cloth = 8200 / 40 = rs 205
cost of 1 metre cloth = rs 205 – 35 = rs 170
cost on 40 metres = 170 x 40 = rs . 6800
profit earned on 40 metres cloth = rs . 8200 – rs . 6800 = rs . 1400
The answer is (d), rs . 1400

Question: if x < y < z and y - x > 5 , where x is an even integer and y and z are odd integers , what is the least possible value s of z - x ? (Hint: The answer is near to (a), (c)).
Options: (a) 6 (b) 7 (c) 8 (d) 9 (e) 10
A: We know the Answer Hints: (a), (c). With the Answer Hints: (a), (c), we will answer the question.
Let's think step by step
We know x < y < z
to find the least possible value for z - x, we need to find the values for z and x that can be closest to each other.
if x is some even number, then what could be minimum possible odd z.
if x is some even number, y - x > 5 ; y > x + 5
minimum value for y = x + 5 + 2 = x + 7
(note : x + 5 is as even + odd = odd and nearest odd greater than x + 5 is x + 5 + 2)
minimum value for z = y + 2 = x + 7 + 2 = x + 9
(note : z = y + 2 because both z and y are odd, difference between two odd numbers is 2)
s = z - x = x + 9 - x = 9
The answer is (d), 9

Question: what is the difference between the c . i . on rs . 6000 for 1 1 / 2 years at 4 % per annum compounded yearly and half - yearly ? (Hint: The answer is near to (e), (d)).
Options: (a) s . 2.04 (b) s . 2.08 (c) s . 2.02 (d) s . 2.83 (e) s . 2.45
A: We know the Answer Hints: (e), d). With the Answer Hints: (e), (d), we will answer the question.
Let's think step by step
c . i . when interest is compounded yearly = [ 6000 * ( 1 + 4 / 100 ) * ( 1 + ( 1 / 2 * 4 ) / 100 ] = 6000 * 26 / 25 * 51 / 50 = rs

. 6364.8

c . i . when interest is compounded half - yearly = [ 6000 * ( 1 + 2 / 100 ) 2 ] = ( 6000 * 51 / 50 * 51 / 50 * 51 / 50 ) = rs . 6367.25

difference = ( 6367.25 - 6364.8 ) = rs . 2.45.

The answer is (e), s . 2.45

Question: the average weight of a , b and c is 45 kg . if the average weight of a and b be 40 kg and that of b and c be 45 kg , then the weight of b is : (Hint: The answer is near to (d), (b)).

Options: (a) 31 kg (b) 32 kg (c) 33 kg (d) 35 kg (e) none of these

A: We know the Answer Hints: (d), (b). With the Answer Hints: (d), (b), we will answer the question.

Let's think step by step

let a , b , c represent their respective weights.

then , we have : a + b + c = ( 45 x 3 ) = 135 . . . ( i )

a + b = ( 40 x 2 ) = 80 . . . ( ii )

b + c = ( 45 x 2 ) = 90 . . . ( iii )

adding ( ii ) and ( iii ), we get : a + 2 b + c = 170 . . . ( iv )

subtracting ( i ) from ( iv ), we get : b = 35.

b' s weight = 35 kg .

The answer is (d), 35 kg

Question: the compound and the simple interests on a certain sum at the same rate of interest for two years are rs . 11730 and rs . 10200 respectively . find the sum (Hint: The answer is near to (e), (c)).

Options: (a) rs . 17037 (b) rs . 17000 (c) rs . 17276 (d) rs . 170287 (e) rs . 171881

A: We know the Answer Hints: (e), (c). With the Answer Hints: (e), (c), we will answer the question.

Let's think step by step

the simple interest for the first year is 10200 / 2 is rs . 5100

and compound interest for first year also is rs . 5100.

the compound interest for second year on rs . 5100 for one year

so rate of the interest = ( 100 * 1530 ) / ( 5100 * 1 ) = 30% p

so p = ( 100 * 10200 ) / ( 30 * 2 ) = rs . 17000

The answer is (b), rs . 17000

You can have as much text here as you want. The main body must be at most 8 pages long. For the final version, one more page can be added. If you want, you can use an appendix like this one.

The \onecolumn command above can be kept in place if you prefer a one-column appendix, or can be removed if you prefer a two-column appendix. Apart from this possible change, the style (font size, spacing, margins, page numbering, etc.) should be kept the same as the main body.

