# OpenReview forum: "Progressive-Hint Prompting Improves Reasoning in Large Language Models"
_ICML.cc/2024/Workshop/AI4MATH — ICML 2024 Workshop AI4MATH Poster_

### Official Review · Reviewer_BDn1 · 2024-06-10

**Rating:** 7
**Confidence:** 4

**Summary:**

The authors present an intuitively simple criterion for selecting the final answer out of several rounds of interaction with the model, each time asking the model slightly differently based on its previous answers. The criterion is to stop when the last two answers are identical. This method is shown to be improving model performance while at the same time reducing the sampling cost compared to the self-consistency method.

**Questions:**

In addition to the question raised in the weakness part, I suggest the authors dig deeper into the concept of double checking. In mathematical problem solving, the reasoning path of double checking can be different from the reasoning path used to derive the answer.  For example, to verify the correctness of a solution of an algebraic equation, one only need to plug the solution into the equation. Even simple things like double checking "21-6=15" with "15+6=21" would be more interesting.

Of course, there are other possible directions worth exploring into, like what does it mean for the hints to be "progressively better" for a general problem. Overall the paper only briefly touches upon many concepts without formalizing any one of them in general.

**Reasons To Accept:**

This paper offers a practical approach by which one can progressively improve the accuracy of the model's reasoning. It is a novel method and has significantly improved the performance on various arithmetic reasoning datasets, particularly through combination with other CoT methods.

The overall quality of this paper is good. The authors conducted extensive experiments to support their method, and reported some interesting phenomena.

This work draws some analogies between LLM reasoning and human problem solving, which could inspire future work.

**Reasons To Reject:**

The main concern I have with this paper is that it does not make clear how PHP can be applied to "aid mathematicians in solving complex mathematical problems", despite stated so in the Broader Impacts section. Some discussion, or better, experiments toward this direction could be included to demonstrate the possibility of applying this method to more general problem settings. For example, what if the answer is not a simple numeric value, but a vector, an algebraic expression, or other abstract mathematical objects?

---

### Official Review · Reviewer_GVVU · 2024-06-11

**Rating:** 8
**Confidence:** 5

**Summary:**

The paper introduces a novel prompting method called Progressive-Hint Prompting (PHP). This method aims to enhance the reasoning abilities of Large Language Models (LLMs) by using previously generated answers as hints to guide subsequent responses progressively. PHP is designed to be orthogonal to existing methods like Chain-of-Thought (CoT) and self-consistency, allowing for easy combination with state-of-the-art techniques to further improve performance.

**Questions:**

Just curious about computational cost, what is the computational overhead associated with implementing PHP compared to standard CoT and self-consistency methods?

**Reasons To Accept:**

originality:
The paper introduces a novel method called Progressive-Hint Prompting (PHP), which enhances the reasoning capabilities of Large Language Models (LLMs). This approach is original in its strategy to iteratively use previously generated answers as hints to guide subsequent responses, mimicking a human-like iterative reasoning process.

quality:
The paper includes detailed ablation studies and baseline comparisons, further strengthening the credibility of the findings. Extensive experimental evaluations on seven benchmarks demonstrate the efficacy of PHP. The results show significant improvements in accuracy, indicating a high-quality and robust validation of the method.

clarity:
The paper is well-organized, with a logical flow that guides the reader through the introduction, methodology, experiments, and conclusions.

significance:
PHP significantly improves the performance of LLMs on reasoning tasks, which is a crucial area of research in natural language processing and artificial intelligence. The demonstrated improvements on benchmarks like SVAMP, GSM8K, AQuA, and MATH underscore the potential of PHP to become a widely adopted technique in the field. By enabling more efficient and accurate reasoning, PHP can be applied to a wide range of practical applications, from education and tutoring systems to advanced AI-driven research tools.

**Reasons To Reject:**

n/a

---

### Official Review · Reviewer_Xxfy · 2024-06-12

**Rating:** 7
**Confidence:** 4

**Summary:**

This paper discussed how to guide LLM to better answering questions by repeatedly providing previous answers given by LLM  to the same question; from the experiment results, it seems that this method helps LLM to converge to the correct answers. Though not clearly mentioned in the original paper, but this experiment may help to understand the inner geometric structures of LMM and is worthy to be introduced to a wider audience.

**Questions:**

1. Have you tried to let 2 intendent LLM talking to each other, discussing the same question and giving each other hints? Then see if the final answer is correct or not?

2. It seems that the accuracy improved over all the dataset by using GPT4 + PHP is 3%; did this happen by chance or there is some deeper reason behind it?

**Reasons To Accept:**

The method proposed in this paper is straightforward but interesting; it seems that the proposed method can guide LLM to find a convergent answer sequence by applying an iterative method which is a common trick in mathematics. This experiment and finding is interesting enough to be introduced.

**Reasons To Reject:**

This paper is good for a poster or even a contributed talk; but it will be better if the authors could provide some theory to support their findings.

---

### Official Review · Reviewer_eu5w · 2024-06-12

**Rating:** 3
**Confidence:** 3

**Summary:**

The authors propose, Progressive-Hint Prompting, a new prompting scheme designed to iteratively improve LLM performance by including the models' previously generated by the model as "hints." The authors claim that Progressive-Hint Prompting improves over baseline prompting schemes. Progressive-Hint Prompting can be further modularly combined with other prompt methods like CoT.

**Questions:**

I included several questions above -- but I would like more details in particular on how the hints were crafted, and would encourage the authors to reconsider whether the full answer can count as a hint. Prompting with actual hints (not the full answer!) could be promising.

**Reasons To Accept:**

The LLM community needs to better understand what kinds of prompts impact behavior, and how we can get the most out of the systems that have been trained. Works which shed light into the impact of different prompting strategies, like this one, are therefore valuable to the broader field.

**Reasons To Reject:**

However, while a deeper understanding of the impact of prompts is important - I have some deep concerns about the validity of this method. It's not clear to me that the prompting method here should be consider as "prompting with hints" -- the model is prompted with its answer. The authors acknowledge that "in this paper, we consider the hint as the answer." But fundamentally, I do not believe that can be referred to as a hint! A hint reveals a snippet of the answer, but not *the* answer.

I am therefore quite skeptical of some of the claims to "humanlike reasoning" that the authors make of their method. While verifying that an answer is correct is a challenge - for humans and models - and prompting a model to verify an answer is a no-doubt valid prompting approach, treating the answer as a hint is different. The hints are also handcrafted? This seems like it would substantially impact scalability. I would be keen to read more details on the nature of the design of these prompts - which humans designed them, the authors?

I may have misunderstood the authors' method and/or intent. Unfortunately, I found the paper to be quite hard to read. Things like "prompt power" were not defined.

---

### Meta-Review · Area_Chair_Zgkz · 2024-06-13

**Recommendation:** Accept (Poster)
**Confidence:** 4

**Metareview:**

This paper is interesting. Because of the results, the fact that having an automatic prompting chain helping an LLM to converge to the correct answer, and the opinion on the three last reviewers, I recommend this paper for acceptation. However, I agree with some of the concerns of reviewer eu5w and in particular about the "humanlike reasoning" given the approach and I can only encourage the authors to take them into account.

---

### Decision · Program_Chairs · 2024-06-13

Accept (Poster)